# CSLP-AE: A Contrastive Split-Latent Permutation Autoencoder Framework for Zero-Shot Electroencephalography Signal Conversion

**Anders Vestergaard Nørskov**     **Alexander Neergaard Zahid**     **Morten Mørup**
Department of Applied Mathematics and Computer Science
Technical University of Denmark
andersxa@gmail.com     {aneol,mmor}@dtu.dk
https://github.com/andersxa/CSLP-AE

## Abstract

Electroencephalography (EEG) is a prominent non-invasive neuroimaging technique providing insights into brain function. Unfortunately, EEG data exhibit a high degree of noise and variability across subjects hampering generalizable signal extraction. Therefore, a key aim in EEG analysis is to extract the underlying neural activation (content) as well as to account for the individual subject variability (style). We hypothesize that the ability to convert EEG signals between tasks and subjects requires the extraction of latent representations accounting for content and style. Inspired by recent advancements in voice conversion technologies, we propose a novel contrastive split-latent permutation autoencoder (CSLP-AE) framework that directly optimizes for EEG conversion. Importantly, the latent representations are guided using contrastive learning to promote the latent splits to explicitly represent subject (style) and task (content). We contrast CSLP-AE to conventional supervised, unsupervised (AE), and self-supervised (contrastive learning) training and find that the proposed approach provides favorable generalizable characterizations of subject and task. Importantly, the procedure also enables zero-shot conversion between unseen subjects. While the present work only considers conversion of EEG, the proposed CSLP-AE provides a general framework for signal conversion and extraction of content (task activation) and style (subject variability) components of general interest for the modeling and analysis of biological signals.

## 1 Introduction

Electroencephalography (EEG) is a non-invasive method for recording brain activity, commonly used in neuroscience research to analyze event-related potentials (ERPs) and gain insights into cognitive processes and brain function [28]. However, EEG signals are often noisy, contain artifacts, and exhibit high sensitivity to subject variability [13, 43], making it challenging to analyze and interpret the data. In particular, the inherent subject variability is well known to confound recovery of the task content of the signal, and a study by Gibson et al. [17] demonstrated that across-subject variation in EEG variability and signal strength was more significant than across-task variation. A major challenge modeling EEG data is thus to remove the intrinsic subject variability in order to recover generalizable patterns of the underlying neural patterns of activation.

Supervised methods explicitly classifying tasks can potentially filter subject variability and recover generalizable patterns of neural activity. However, explicitly disentangling subject and task content in EEG signals is valuable not only for classifying task content but also for characterizing the subject variability which ultimately can provide biomarkers of individual variability. As such, rather than

37th Conference on Neural Information Processing Systems (NeurIPS 2023).

focusing only on the experimental effects and considering inter-subject variability as noise it should be treated as an important signal enabling the understanding of individual differences [24, 55].

Deep learning-based feature learning has become prevalent in EEG data analysis. As such, auto-encoders have shown promise in learning transferable and feature-rich representations [36, 60, 70] and have been applied to various downstream tasks, including brain-computer interfacing (BCI) [35, 44, 78], clinical epilepsy and dementia detection [19, 38, 77], sleep stage classification [34, 63], emotion recognition [26, 61], affective state detection [53, 74] and monitoring mental workload/fatigue [75, 76] with promising results. For a systematic review on deep learning-based EEG analysis, see e.g. Roy et al. [52].

The task of disentangling content (signal) from style (individual variability) is a well-known aim in voice conversion technologies. A study on speech representation learning achieved promising results in disentangling speaker and speech content using speaker-conditioned auto-encoders [9], while Chou et al. [10] proposed using instance normalization to enforce speaker and content separation in the latent space. In Qian et al. [48] zero-shot voice conversion was proposed using a simple autoencoder conditioned on a pretrained speaker embedding model and exploring bottleneck constriction to obtain content and style disentanglement with good results. Wu and Lee [71] and Wu et al. [72] disregarded the pretrained speaker embedding model by generating content latents based on a combination of instance normalization and vector quantization of an encoded signal, while the speaker latents were generated based on the difference between content codes and the encoded input signal.

Disentangling subject and task content has been shown to enhance model generalization in emotional recognition and speech processing tasks [4, 50, 57]. Bollens et al. [4] used two explicit latent spaces in a factorized hierarchical variational autoencoder (FHVAE) to model high-level and low-level features of EEG data and found that the model was able to disentangle subject and task content. They also found that high-level features were more subject-specific and low-level features were more task-specific. Rayatdoost et al. [50] and Özdenizci et al. [42] explored adversarial training to promote latent representations that were subject invariant. Recently, self-supervised learning has gained substantial attention due to its strong performance in representation learning enabling deep learning models to efficiently extract compact representations useful for downstream tasks. This includes the use of (pre-)training on auxiliary tasks [62] as well as contrastive learning methodologies guiding the latent representations [25]. Shen et al. [57] used contrastive learning between EEG signal representations of the same task and different subjects to learn subject-invariant representations and found that the learned representations were more robust to subject variability and improved generalizability. For a survey of self-supervised learning in the context of medical imaging, see also [58].

Inspired by recent advances in voice conversion, we propose a novel contrastive split-latent permutation autoencoder (CSLP-AE) framework that directly optimizes for EEG conversion. In particular, *we hypothesize that the auxiliary task of optimizing EEG signal conversion between tasks and subjects requires the learning of latent representations explicitly accounting for content and style.* We further use contrastive learning to guide the latent splits to respectively represent subject (style) and task (content). The evaluation of the proposed method is conducted on a recent, standardized ERP dataset, ERP Core [28], which includes data from the same subjects across a wide range of standardized paradigms making it especially suitable for signal conversion across multiple tasks and subjects. We contrast CSLP-AE to conventional supervised, unsupervised (AE [23]), and self-supervised training (contrastive learning [7, 41, 49, 59, 69]).

## 2 Methodology

The aim of this paper is to develop a modeling framework for performing generalizable (i.e., zero-shot) conversion of EEG data considering unseen subjects. The procedure should enable conversion of EEG from one subject to another as well as one task to another task. In this context, "tasks" refer to the specific ERP components present in the EEG data, such as face or car perception, word judgement of related or unrelated word pairs, perception of standard or deviant auditory stimuli, etc. [28].

The standard autoencoder (AE) model consists of an encoder, denoted as $E_\theta(\boldsymbol{X}) : \mathcal{X} \rightarrow \mathcal{Z}$, and a decoder, denoted as $D_\phi(\boldsymbol{Z}) : \mathcal{Z} \rightarrow \mathcal{X}$. The encoder maps the input data to a latent space, while the decoder reconstructs the input from the latent space. The encoder and decoder are parameterized by $\theta$ and $\phi$ respectively.

To enable the model to perform conversion, it needs to be conditioned on the target subject and/or task. The latent space appears to be the suitable place for conditioning, as it is a compact representation often referred to as the bottleneck of the model. However, since the latent space is shared across subject and task representations, partitioning it into specific subject and task streams is non-trivial.

To address this challenge, a split-latent space is explored, which explicitly divides the latent space into subject and task disentangled representations. This is achieved by introducing a split in the model design within the encoder. The split-latents can be obtained by encoding the input data as follows: $E_\theta(\boldsymbol{X}) = (\boldsymbol{z}^{(\mathcal{S})}, \boldsymbol{z}^{(\mathcal{T})})$, where $\boldsymbol{z}^{(\mathcal{S})}$ represents the subject latent and $\boldsymbol{z}^{(\mathcal{T})}$ represents the task latent, such that $E_\theta(\boldsymbol{X}) : \mathcal{X} \to (\mathcal{S}, \mathcal{T})$. Note that the $(\mathcal{S})$ and $(\mathcal{T})$ denominations are not inherent from the model architecture but are necessary distinctions for use in the model loss functions. These split-latents can then be joined and decoded using a similar split within the decoder. By feeding the split-latents into the decoder, the input can be reconstructed as $\hat{\boldsymbol{X}} = D_\phi(\boldsymbol{z}^{(\mathcal{S})}, \boldsymbol{z}^{(\mathcal{T})})$, such that $D_\phi(\boldsymbol{z}^{(\mathcal{S})}, \boldsymbol{z}^{(\mathcal{T})}) : (\mathcal{S}, \mathcal{T}) \to \mathcal{X}$. The concept of split-latent space has been explored by researchers within speech [48] and in the EEG domain [4] using separate encoders for each latent space. We presently employ a shared encoder to reduce the number of parameters used. However, separate encoders - or even pre-trained encoders, specifically, on subjects - could help kick-start the training process or work as conditioning with frozen weights. We leave this to further studies.

While voice conversion [48, 68] and other voice synthesis problems [40, 56] based on autoencoders generally use models with expansive receptive fields, e.g. as in WaveNet [40], other studies have found similar performance in voice conversion using simpler architectures, such as CNNs [10, 27, 33]. Voice conversion is usually done over a large number of samples with exceptionally high sampling rate compared to EEG data. ERP Core uses a sampling frequency of 256 Hz (downsampled from 1024 Hz) over an epoch window of 1 second, which only yields a time resolution of 256 samples. Therefore, a large receptive field is unnecessary, and strided convolution to reduce time-resolution is used instead. The proposed EEG auto-encoder model with split latent space is illustrated in Figure 1.

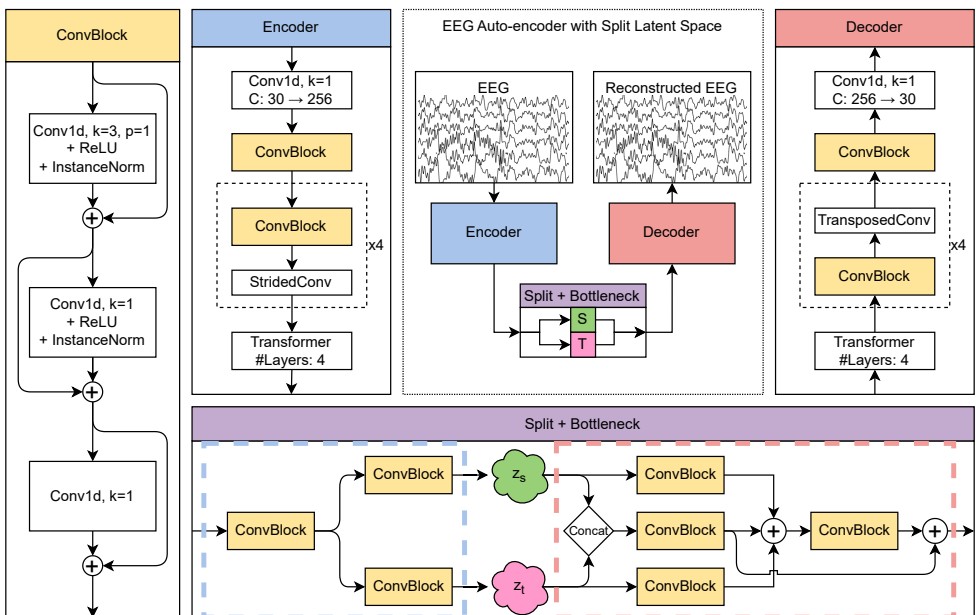

**Figure 1:** The proposed EEG auto-encoder model with split latent space. The encoder and decoder are mirrored deep convolutional neural networks using one-dimensional convolutions. Each part of the auto-encoder consists of ConvBlocks which are made up of three convolutions with residual connections (as in He et al. [22]), rectified linear unit (ReLU) activation [15] and instance normalization [64] (similar to Chou et al. [10]). The encoder applies a ConvBlock together with a strided convolution (stride=2), to reduce the time-resolution by half, four times. The decoder is mirrored and uses transposed convolutions (fractionally strided, stride=½) to upscale the time-resolution by a factor of two with each block. Both the encoder and decoder models use a transformer [66] with four layers on each side of the bottleneck to confer attention. Finally, on the encoder side, a split is made into subject and task latent spaces. The decoder takes these split-latents as inputs and joins them again in the bottleneck in order to reconstruct the input. $k$ is the kernel size and $p$ is the padding on both sides.

## 2.1 Split-latent permutation

To guide the autoencoder in Figure 1 in disentangling task and subject we propose the use of latent-permutation, which is a self-supervised approach that guides the latent space by ensuring consistency in the subject and/or task encodings between permutations. This can be seen as a direct loss relating to the conversion method described in this section.

To achieve zero-shot conversion the respective subject and task information need to be extracted from the input data, and with an explicit split of the latent space, the conversion becomes straightforward and practical. Depending on the desired conversion task, such as converting from subject $U$ to subject $V$ or from task $M$ to task $N$, the corresponding split-latents are simply swapped with that of the target subject or task. This conversion method is illustrated in Figure 2a.

Given a pair of input samples $(\boldsymbol{X}_i^a, \boldsymbol{X}_i^b)$, where $i$ is the batch index, the two pairs of split-latents are defined from $E_\theta$ splitting the latent space in two parts yielding $(\boldsymbol{z}_i^{(\mathcal{S},a)}, \boldsymbol{z}_i^{(\mathcal{T},a)})$ and $(\boldsymbol{z}_i^{(\mathcal{S},b)}, \boldsymbol{z}_i^{(\mathcal{T},b)})$. A latent permutation is performed which swaps two of the latents in a given latent space $\mathcal{L} \in \{\mathcal{S}, \mathcal{T}\}$ before reconstruction. A comprehensive glossary of symbols and abbreviations is provided in the appendix.

Consider the latent space $\mathcal{L} = \mathcal{T}$. The pair of input samples $(\boldsymbol{X}_i^a, \boldsymbol{X}_i^b)$ are sampled to both belong to the same task $t_i$. The task latents are swapped between the pairs such that the reconstructed EEG data decoded by $D_\phi$ becomes

$$E_\theta(\boldsymbol{X}_i^a) = (\boldsymbol{z}_i^{(\mathcal{S},a)}, \boldsymbol{z}_i^{(\mathcal{T},a)}), \qquad \hat{\boldsymbol{X}}_i^{(\mathcal{T},a)} = D_\phi(\boldsymbol{z}_i^{(\mathcal{S},a)}, \boldsymbol{z}_i^{(\mathcal{T},b)}) \qquad (1)$$

$$E_\theta(\boldsymbol{X}_i^b) = (\boldsymbol{z}_i^{(\mathcal{S},b)}, \boldsymbol{z}_i^{(\mathcal{T},b)}), \qquad \hat{\boldsymbol{X}}_i^{(\mathcal{T},b)} = D_\phi(\boldsymbol{z}_i^{(\mathcal{S},b)}, \boldsymbol{z}_i^{(\mathcal{T},a)}) \qquad (2)$$

here colorized according to their corresponding input sample $\boldsymbol{X}_i^a$ or $\boldsymbol{X}_i^b$. This is the same-task latent-permutation since pairs belong to the same task class. A corresponding swap can be done for the subject latent space $\mathcal{S}$ with pairs belonging to the same subject, $s_i$. The task and subject latent space permutations are illustrated in Figure 2b and Figure 2c respectively.

The latent-permutation loss is defined as the sum over the pair of reconstruction losses between the two samples and their corresponding reconstructions with split-latents from latent space $\mathcal{L}$ swapped:

$$L_{LP}(\mathcal{L}; \boldsymbol{X}_i^a, \boldsymbol{X}_i^b) = \frac{1}{N} \sum_{i=1}^N \left( \|\boldsymbol{X}_i^a - \hat{\boldsymbol{X}}_i^{(\mathcal{L},a)}\|_2^2 + \|\boldsymbol{X}_i^b - \hat{\boldsymbol{X}}_i^{(\mathcal{L},b)}\|_2^2 \right) \qquad (3)$$

Consider the scenario where $\mathcal{L} = \mathcal{T}$ again using the reconstructions from Eq. (1). In Eq. (3) the $L_2$-norm is calculated between the input data $\boldsymbol{X}_i^a$ and the reconstruction of the latent-permutation $(\boldsymbol{z}_i^{(\mathcal{S},a)}, \boldsymbol{z}_i^{(\mathcal{T},b)})$. According to the smoothness meta-prior proposed by Bengio et al. [2] a pair of task latents should be invariant to local perturbations in the input. Seen in the context of latent-permutation, such a local perturbation could be equivalent to two task representations of the same class $t_i$ from different input samples. When the latent space is locally smooth (i.e. encoder is consistent) then this term approximates the autoencoder reconstruction loss, i.e. if $\boldsymbol{z}_i^{(\mathcal{T},a)} \approx \boldsymbol{z}_i^{(\mathcal{T},b)}$ then $D_\phi(\boldsymbol{z}_i^{(\mathcal{S},a)}, \boldsymbol{z}_i^{(\mathcal{T},a)}) \approx D_\phi(\boldsymbol{z}_i^{(\mathcal{S},a)}, \boldsymbol{z}_i^{(\mathcal{T},b)})$ where $D_\phi(\boldsymbol{z}_i^{(\mathcal{S},a)}, \boldsymbol{z}_i^{(\mathcal{T},a)})$ is the standard reconstruction when both $E_\theta$ and $D_\phi$ are consistent. This is expanded upon in the supplementary material.

Notably, such autoencoding enables flow of structural information specific for the given input sample and its reconstruction beyond subject and task content. In the supplementary material, we explore a setup where the output sample is from a different subject *and* task than the input sample.

In contrast to AutoVC [48], which achieves disentanglement by adding an explicit speaker latent and reducing the capacity of the content encoder through a smaller latent dimension, the proposed split-latent permutation directly optimizes for conversion. It is not guaranteed that it will explicitly separate task and subject information in their respective spaces. When the capacities of the latents are sufficiently large, both subject and task content can potentially be encoded together, and the decoder can learn to extract the corresponding subject and task content from these identically encoded splits. To address this issue and avoid relying solely on disentanglement achieved through careful bottleneck tuning, we explore contrastive learning to suffice the smoothness criteria, disentangle and *specialize* each latent space. Limiting the capacity of the latent space is explored in the supplementary material.

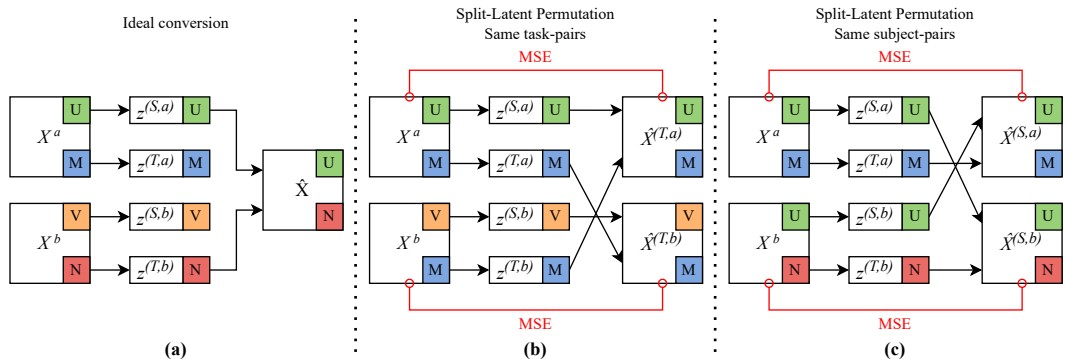

**Figure 2: (a)**: Split-latent conversion example. Two samples $X^a$ and $X^b$, where $X^a$ corresponds to subject $U$ and task $M$ and $X^b$ corresponds to subject $V$ and task $N$ respectively, are encoded yielding a pair of split-latents for each sample. In this example, to convert from task $M$ to task $N$, the subject ($U$) latent is kept while the task ($M$) latent is simply swapped with the corresponding latent from the other sample which encodes task $N$. The split-latents are then decoded to ideally obtain a sample from the data distribution given subject $U$ and task $N$. This is the ideal case where the split-latents are perfectly disentangled and independent. **(b)** and **(c)**: Illustration of the object class-dependant swap. The input samples are encoded yielding their corresponding split-latents ($z^{(S,a)}$, $z^{(S,b)}$, $z^{(T,a)}$, $z^{(T,b)}$). Depending on the object class (tasks in **(b)**, subjects in **(c)**), the split-latents are swapped between the two pairs of latents. These are then decoded yielding the reconstructed EEG data. The same-task pairs both have blue (M) task-latents since these are intentionally sampled from the same class. The intention of the latent-permutation method is expressed when the model is consistent in its latent representation of the class, and when the split latent spaces are sufficiently disentangled in the ideal case. In such cases the swap will have negligible impact on the conversion, i.e. if they encode the same (subject or task) information.

## 2.2 Contrastive Learning

Contrastive learning is a self-supervised learning method which aims to learn a representation of the data by maximizing the similarity between positive pairs and minimizing it between negative pairs [7, 8, 21] achieving local smoothness around classes. We apply contrastive learning to each split-latent space. This is done by utilizing a batch construction technique to sample pairs for both subjects and tasks separately, and then applying the contrastive loss to each split-latent space according to denomination, thus *specializing* the given latent space for either subject or task embeddings.

Specifically, we consider the multi-class $N$-pair loss [59] which is a deep metric learning method [69] utilizing a special batch construction technique. The batch construction technique involves sampling two samples from each class, and then constructing $K$ pairs of samples from the batch. This batch construction is similarly required for the latent-permutation in Section 2.1. We use the batch construction method combined with the InfoNCE generalization from Oord et al. [41] with the $\tau$ temperature parameter as in Chen et al. [7]. The full generalization is equivalent to the CLIP-loss from Radford et al. [49] in which we minimize the symmetric cross-entropy loss of the temperature-scaled similarity matrix based on the NT-Xent loss [7]. Let $\boldsymbol{Z}^A \in \mathbb{R}^{C \times K}$ and $\boldsymbol{Z}^B \in \mathbb{R}^{C \times K}$ be latent representation matrices of the $K$ pairs of samples, then $L_{\text{NT-Xent}}$ and $L_{\text{CLIP}}$ are defined as

$$L_{\text{NT-Xent}}(\mathcal{L}; \boldsymbol{Z}', \boldsymbol{Z}'', k) = -\log \frac{\exp(\text{sim}(\boldsymbol{z}'_k, \boldsymbol{z}''_k)/\tau)}{\sum_{i=1}^{K} \mathbb{1}_{[i \neq k]} \exp(\text{sim}(\boldsymbol{z}'_k, \boldsymbol{z}''_i)/\tau)} \tag{4}$$

$$L_{\text{CLIP}}(\mathcal{L}; \boldsymbol{Z}^A, \boldsymbol{Z}^B) = \frac{1}{K} \sum_{k=1}^{K} \left( L_{\text{NT-Xent}}(\mathcal{L}; \boldsymbol{Z}^A, \boldsymbol{Z}^B, k) + L_{\text{NT-Xent}}(\mathcal{L}; \boldsymbol{Z}^B, \boldsymbol{Z}^A, k) \right) \tag{5}$$

where $\mathbb{1}_{[c]} \in \{0, 1\}$ is the indicator function yielding 1 iff the condition $c$ holds true. $\text{sim}(\boldsymbol{z}'_i, \boldsymbol{z}''_k)$ is a similarity metric. $\mathcal{L}$ denotes from which latent space the pairs have corresponding labels, e.g. for the task latent space $\mathcal{T}$, the $k$'th pair has the same task $t_k$ which is different from the other tasks in the same batch, i.e. $t_1 \neq t_2 \neq \ldots \neq t_K$. $L_{\text{CLIP}}(\mathcal{T}; \cdot, \cdot)$ therefore is a contrastive loss across tasks, while $L_{\text{CLIP}}(\mathcal{S}; \cdot, \cdot)$ is across subjects.

# 3   Experimental Setup

**Data**   We consider the ERP Core dataset[1] from Kappenman et al. [28] providing a standardized ERP dataset containing data from 40 subjects across six different tasks based on seven widely used ERP components: N170 (Face Perception Paradigm), MMN (Mismatch Negativity, Passive Auditory Oddball Paradigm), N2pc (Simple Visual Search Paradigm), N400 (Word Pair Judgement Paradigm), P3 (Active Visual Oddball Paradigm), and LRP and ERN (Lateralized Readiness Potential and Error-related Negativity, Flankers Paradigm). We only consider data processed by Script #1 up until ICA preparation.[2] For more information on the data and paradigms, see Kappenman et al. [28].

Only the epoch windows around the time-locking event as described in Kappenman et al. [28] are used as epochs, therefore, for each paradigm there are two available "tasks" each with a different resulting ERP. The two classes assigned for each event per paradigm are: N170; faces/cars, MMN; deviants/standards, N2pc; contralateral/ipsilateral, N400; unrelated/related, P3; rare/frequent, and ERN and LRP; incorrect/correct. The dataset was split across subjects into a training set of 70%, an evaluation set of 10%, and a test set of 20% of the subjects respectively. The exact splits are available in the supplementary material.

The ERP Core dataset is predominantly time-locked with data centered around either stimulus or response[3]. We further evaluate the models "as is" on two other modalities of EEG data from PhysioNet [18] and with already established state of the art model results: the EEG Motor Movement/Imagery Dataset (EEGMMI) [54] and the Sleep-EDF Expanded (SleepEDFx) database [29]. The EEGMMI dataset is cue time-locked and consists of recordings from 109 subjects performing various motor imagery (MI) tasks. We applied our model to the standard L/R/0/F MI task using 3s epoch windows, closely following the approach of Wang et al. [67]. The SleepEDFx is included to explicitly probe the model integrity on data that is not time-locked to an external stimulus and contains 153 polysomnography studies from 78 subjects. Given its limited EEG channels, we adapted our methodology by considering a single EEG channel and applied a short-time Fourier transform to fit the data to the same setup as used for ERP Core. We maintained the conventional 30s time series windows commonly used in sleep stage literature. Details on the data preprocessing and model integration for these datasets can be found in the supplementary materials.

**Model comparisons**   The proposed CLSP-AE approach is systematically compared against a mix of learning strategies comprising conventional AE-based representation learning, contrastive learning and representations obtained by supervised learning. To optimally compare these learning strategies all models are based on the same model structure given in Figure 1. Models will be denoted by which losses they are trained on, or which external methods they use. Here CSLP-AE will denote the contrastive split-latent permutation over both subjects and latents ($L_{LP}(\mathcal{S}; \cdot, \cdot)$, $L_{LP}(\mathcal{T}; \cdot, \cdot)$, $L_{\text{CLIP}}(\mathcal{S}; \cdot, \cdot)$ and $L_{\text{CLIP}}(\mathcal{T}; \cdot, \cdot)$), SLP-AE the corresponding model without the contrastive loss components ($L_{LP}(\mathcal{S}; \cdot, \cdot)$, and $L_{LP}(\mathcal{T}; \cdot, \cdot)$), CL the contrastive loss in both of the split latent spaces ($L_{\text{CLIP}}(\mathcal{S}; \cdot, \cdot)$ and $L_{\text{CLIP}}(\mathcal{T}; \cdot, \cdot)$). Cosine similarity will be used as the similarity metric in the contrastive loss. AE will denote the standard auto-encoder with mean-squared error reconstruction loss on the reconstructions of non-permuted split-latents. C-AE will denote the combination of reconstruction loss and contrastive learning, such that contrastive learning is applied in both spaces *and* the standard autoencoder reconstruction loss is applied to non-permuted reconstructions. CE will denote the supervised learning cross-entropy loss jointly trained in the subject and task latent spaces using the corresponding true labels in a supervised manner. Similarly, CE(t) will denote cross-entropy only trained on the task labels in the task latent space. This is to substitute and compare with a supervised deep learning model, contrary to the self-supervised methods described here. For completeness, we also included the common spatial pattern (CSP) method [3, 32] using the multi-class generalization from Grosse-Wentrup and Buss [20] which is a supervised method for extracting discriminative features from EEG data. These features are then used to map unseen data into the same "CSP space". For the EEGMMI and SleepEDFx datasets we respectively compared the task and sleep stage classification performance to [12, 67, 73] and [14, 39, 45–47, 79].

---

[1]ERP Core info: `https://erpinfo.org/erp-core`. Data available at: `https://osf.io/thsqg/`

[2]See the full ERP Core procedure here: `https://github.com/lucklab/ERP_CORE/blob/master/ERN/ERN%20Analysis%20Procedures.pdf`

[3]Only the ERN and LRP paradigms are response time-locked

The total loss for models with multiple losses is the sum of losses with equal weighting. This is further detailed in the supplementary materials and loss curves are shown in the appendix.

Hyperparameters were chosen based on evaluation during development, and the most critical hyperparameters (the number of blocks and the size of the latent space) were verified on the evaluation set in a grid search. See supplementary material for details on the grid search across both latent dimension and number of blocks. The test set was used only for final evaluation.

**Subject and task characterization**   Non-linear classifiers were trained to classify the subject and task labels of split-latents from the test set to quantify the disentanglement and generalization of the latent spaces. This was performed as two five (5)-fold cross-validations (CVs) over the subject and task latents respectively stratified on the subject and task labels on the test data (unseen subjects). Since there is high class-imbalance in the task labels, undersampling was performed for each class to match the number of samples in the least represented class. The undersampling was performed on the training split of each fold, and the test split was left untouched. Balanced accuracy [5] was used due to the high class-imbalance. The subject CV was used to evaluate the subject classification accuracy (S.acc%) and task-on-subject classification accuracy (T⊢S.acc%), while the task CV was used to evaluate the task classification accuracy (T.acc%) and subject-on-task classification accuracy (S⊢T.acc%). The task-on-subject classification accuracy was evaluated by training a classifier on the subject latents to predict the task of the corresponding input sample, and vice versa for the subject-on-task classification accuracy.

For each fold, an end-to-end tree boosted system (XGBoost) [6] was trained on the training split, and subsequently evaluated on the test split. Finally, the results were averaged over the five (5) folds. All classifications were performed on a single-trial level. A K-nearest neighbors (KNN) classifier and an Extra Trees classifier [16] were also trained and evaluated. See supplementary material for these.

**ERPs from zero-shot EEG conversions**   An ERP conversion loss was measured on the test set. First, a ground-truth ERP was found for each subject and for each ERP component (task) by averaging over all samples belonging to the same subject and task only in the EEG channel respective to which Kappenman et al. [28] found the given ERP component most prominent. Let $\hat{x}^{\text{ERP}}_{(s,t)}$ denote such an ERP for subject $s$ and task $t$. If disentanglement was successful then the conversion method described in Section 2.1 and illustrated in Figure 2 should be able to reconstruct the ERP from a given subject and task latent pair. Thereby it should be possible to sample an amount of subject and task latent pairs encoded on the test set, and convert the ERP from these pairs. Let S.s. and D.s. denote sampling *task latents* from the same or different target subject $\sigma$ respectively, and let S.t. and D.t. denote sampling *subject latents* from the same or different target task $\gamma$ respectively. For a conversion to be valid subject latents must all come from samples with target subject $s_k = \sigma$ and task latents from samples with $t_k = \gamma$. S.s., D.s., S.t., and D.t. are then used as additional conditions for sampling. We then consider all combinations (S.s., S.t.), (D.s., D.t.), (D.s., S.t.), (S.s., D.t.) for different conversion abstractions. $N$ pairs of subject and task latents corresponding to the targets are drawn using the specific considered combination of conditions. The EEG is reconstructed for the $k$'th sample to obtain $\hat{x}^{(\sigma,\gamma)}_k \in \mathbb{R}^T$ where $T$ is the number of time-samples. The converted ERP (C-ERP) is measured by averaging over each of these converted EEG signals: $\hat{x}^{\text{C-ERP}}_{(\sigma,\gamma)} = \frac{1}{N} \sum_{k=1}^{N} \hat{x}^{(\sigma,\gamma)}_k$. The ERP conversion loss is calculated as the mean squared error (MSE) between the converted ERP and the per-subject per-task ERP: $L_{\text{C-ERP-MSE}}(\sigma,\gamma) = \frac{1}{T}\|\hat{x}^{\text{ERP}}_{(\sigma,\gamma)} - \hat{x}^{\text{C-ERP}}_{(\sigma,\gamma)}\|^2_2$. An illustration of this procedure and examples of these ERPs are available in supplementary material.

An ERP was measured for each subject and task combination and was compared with the corresponding ERP from converted EEGs. The reported ERP conversion MSE is the average over all target subject and target task combinations. The number of samples ($N$) was set to 2000 for all methods. See supplementary material for an analysis of how this number affects the ERP conversion MSE.

A more detailed summary of the data, pre-processing, hyperparameters and training for all models, architectures and methods can be found in the supplementary material.

# 4 Results and Discussion

Table 1 shows the results of the task and subject classification as well as EEG conversion. The stand-alone CL, CE and CE(t) models do not have a decoder, and therefore do not have a reconstruction loss. The standard error of the mean is reported for all classification accuracies and ERP conversion MSEs over the five ($n = 5$) repeats of each model.

**Table 1:** Single-trial balanced subject classification accuracy (S.acc%), task-on-subject classification accuracy (T⊢S.acc%), task classification accuracy (T.acc%), subject-on-task classification accuracy (S⊢T.acc%), and zero-shot same-subject same-task (S.s., S.t.), different-subject different-task (D.s, D.t.), different-subject same-task (D.s., S.t.), and same-subject different-task (S.s., D.t.) ERP conversion MSE. All ERP conversion MSE values have scales of $10^{-11}\text{V}^2$. Confusion matrices are provided in the supplementary material.

| Model | S.acc% | T⊢S.acc% | T.acc% | S⊢T.acc% | (S.s., S.t.) | (D.s, D.t.) | (D.s., S.t.) | (S.s., D.t.) |
|---|---|---|---|---|---|---|---|---|
| CSLP-AE | **80.32 ± 0.28** | 45.41 ± 0.37 | **48.48 ± 0.34** | 79.64 ± 0.37 | 4.21 ± 0.12 | 20.06 ± 0.10 | **5.80 ± 0.15** | 6.65 ± 0.23 |
| SLP-AE | 74.63 ± 0.74 | 47.23 ± 0.31 | 47.00 ± 0.32 | 74.70 ± 0.73 | 3.82 ± 0.04 | **19.92 ± 0.10** | 6.12 ± 0.09 | **5.02 ± 0.08** |
| C-AE | 79.42 ± 0.48 | 37.34 ± 0.45 | 46.59 ± 0.23 | 73.27 ± 0.25 | 4.28 ± 0.06 | 20.28 ± 0.07 | 11.33 ± 0.47 | 10.64 ± 0.30 |
| AE | 60.68 ± 0.16 | 31.62 ± 0.27 | 31.43 ± 0.28 | 61.08 ± 0.38 | **3.54 ± 0.12** | 20.82 ± 0.07 | 11.20 ± 0.32 | 10.74 ± 0.48 |
| CL | 78.82 ± 0.46 | 37.65 ± 0.54 | 45.36 ± 0.37 | 71.70 ± 0.55 | - | - | - | - |
| CE | 79.25 ± 0.37 | 35.52 ± 0.38 | 45.22 ± 0.23 | 64.73 ± 0.44 | - | - | - | - |
| CE(t) | - | - | 45.80 ± 0.24 | 44.27 ± 0.59 | - | - | - | - |
| CSP | - | - | 35.22 ± 0.11 | 69.89 ± 0.10 | - | - | - | - |

From the results on ERP Core (Table 1), the best subject and task classifications were obtained using the proposed CSLP-AE whereas the second best performant models for subject classification and task classification were C-AE and SLP-AE respectively. We further observe a substantial performance increase in the subject and task classification of the SLP-AE when compared to the conventional AE whereas SLP-AE even provides higher task accuracies than conventional supervised training (i.e., CL, CE, CE(t), and CSP). Importantly, AE and C-AE exhibit poor conversion performance and can only reconstruct good ERPs in the standard autoencoder regime considering same subject and same task (S.s., S.t.). We observe that the SLP-AE and CSLP-AE both perform well in conversion when either the task or subject latents come from other samples (D.s., S.t.) and (S.s., D.t.) achieving reconstruction errors similar to (S.s., S.t.). Examples of converted ERPs from (D.s., S.t.) and (S.s., D.t.) are shown in Figure 3b.

Experiments on the EEGMMI dataset (Table 2) showed similar results with the CSLP-AE model outperforming both the C-AE and SLP-AE models, and achieving on par performance with the current state-of-the-art model: CSLP-AE task accuracy of 64.28±0.16% compared to 65.07% achieved by EEGNet[67]. However, the SLP-AE model performed considerably worse than the CSLP-AE and C-AE models which performed similarly on the SleepEDFx dataset. Split-latent permutation, therefore, does not seem to increase task classification accuracy on non-time-locked data.

**Table 2:** Task classification accuracy on PhysioNet [18] EEG Motor Movement/Imagery Dataset [54] (EEGMMI) and Sleep-EDF Expanded Dataset [29] (SleepEDFx)

| Model | EEGMMI | SleepEDFx |
|---|---|---|
| CSLP-AE (ours) | 64.28 ± 0.16 | 75.16 ± 0.95 |
| C-AE (ours) | 61.89 ± 0.41 | 75.16 ± 0.86 |
| SLP-AE (ours) | 57.93 ± 0.56 | 70.59 ± 1.18 |
| EEGNet (Wang et al. [67]) | **65.07** | - |
| f-CTrans (Xie et al. [73]) | 64.22 | - |
| CNN (Dose et al. [12]) | 58.59 | - |
| XSleepNet2 (Phan et al. [46]) | - | **84.0** |
| Zhu et al. [79] | - | 82.8 |
| SeqSleepNet (Phan et al. [45]) | - | 82.6 |
| SleepTransformer (Phan et al. [47]) | - | 81.4 |
| AttnSleep (Eldele et al. [14]) | - | 81.3 |
| SleepEEGNet (Mousavi et al. [39]) | - | 80.0 |

The CSLP-AE model achieved a sleep stage classification accuracy of 75.16±0.96% which is notably lower than the state-of-the-art model XSleepNet2 [46] with 84.0% accuracy. With an accuracy of 75.16±0.96%, the model demonstrates a capability beyond mere chance, effectively characterizing sleep stages outside the constraints of the time-locked paradigm. We presently restricted the model to Fourier-compressed representations of the data but we expect performance could be increased using encoders with larger receptive fields such as WaveNet [9, 40, 56] on the raw EEG waveform data.

$t$-distributed stochastic neighbor embedding [1, 31, 65] ($t$-SNE) plots are provided in Figure 3a for the SLP-AE, C-AE and CSLP-AE models on the ERP Core dataset (further details of the $t$-SNE and additional plots are provided in the supplementary). From the figure we observe that the SLP-AE

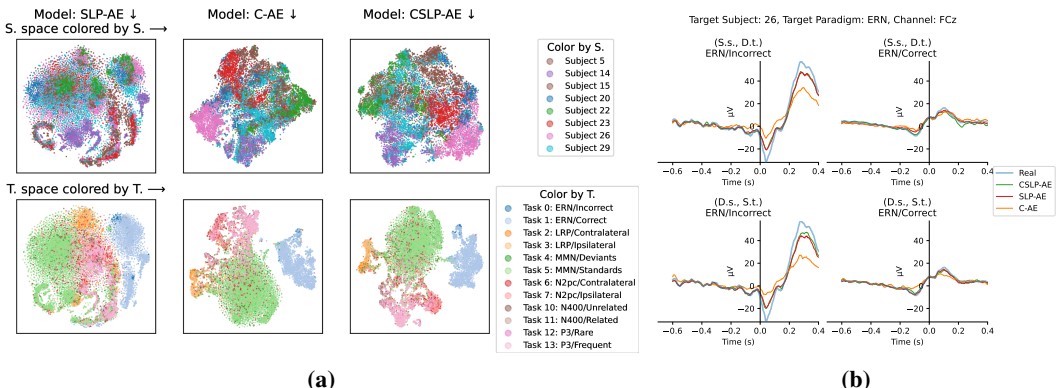

**(a)**                                                                                     **(b)**

**Figure 3: (a)** $t$-SNE plots of split-latents as encoded on the test set (unseen subjects) of the ERP Core dataset, colored by true labels. Rows show *subject* and *task* latent spaces, respectively, while columns indicate model type (SLP-AE, C-AE, and CSLP-AE respectively). For task latent space colorized by subject, and vice versa, see supplementary material. **(b)** Converted ERPs from the same three models for a random target subject and target paradigm. All latents used in conversion were from unseen subjects on the test set, i.e. unseen to unseen conversion. The FCz channel was chosen according to which channel Kappenman et al. [28] found to most prominently show the ERP of the given paradigm. For more conversion examples see supplementary material.

trained subject and task spaces are topologically similar. We further observe that the addition of the split-latent permutation loss to the contrastive learning model had little effect on the topology of the task latent space but some effect on the subject latent space. As the observed topology of the latent space did not undergo significant changes, the improvement in conversion performance can primarily be attributed to the capabilities of the decoder and the information flow within it to implicitly account for the permutation-invariance. This points to the importance of accounting for *structural encoding*.

The different ERP conversion abstractions described in Section 3 are increasingly more difficult to convert from. (S.s., S.t.) allows structural information through both latent spaces, while (D.s., S.t.) and (S.s., D.t.) allow structural information through one latent space since one of the latents will come from the same input sample. (D.s., D.t.) is a hard problem requiring conversion with no retention of any structural information from the specific sample for the decoder to rely on, i.e. conversion must be done using only abstract subject and task embeddings from which structure must arise.

Models trained with latent-permutation learns this structural encoding in the latent spaces since the decoder can rely on at least one of the latent spaces in the latent-swap procedure (Figure 2) to provide the structural information of the EEG signal. This structural information, although indistinguishable in different latent spaces, coincidentally is highly correlated with both subject and task content of the signal. This structural encoding is most notable in the SLP-AE latent space as shown in Figure 3a allowing the classifier to perform well on all classification tasks. However, the SLP-AE model obtains worse performance on subject classification compared to the other deep learning models. Interestingly, it had identical performance of subject classification on either subject or task latents, and similarly for task classification (S.acc%≈S⊢T.acc% and T.acc%≈T⊢S.acc% for SLP-AE in Table 1), which further accentuates their topological similarities seen in Figure 3a.

Contrastive learning in the latent space itself has nothing to do with decoding the structure of the data for reconstruction. The encoder is simply trained to learn an embedding of the subject and task content of the signal. The decoder must, therefore, do the heavy work of extracting this structural information itself. This exposes a property which is applicable to the latent-permutation method but not contrastive learning. When the stand-alone latent-permutation method is used (SLP-AE), the decoder is allowed a reliability in the latent spaces.

A lot of the EEG signal is structural information, therefore, there might be more to gain from minimizing the permutation-invariance to structural information rather than encoding the subject and task content directly. Thereby, the decoder can learn a permutation-invariant structural encoding of the signal in both latent spaces, which allows the decoder to rely on this information irrespective of the permutation or swap - since only one latent space (see Figure 2) is swapped at a time. Thus, the latent-permutation only trained model (SLP-AE) does not learn explicitly disentangled representations of the subject and task content of the signal, but rather a duplicated latent space permutation-invariant

structural encoding which is highly correlated with both subject and task content of the signal. This interestingly is also the goal of the standard autoencoder. The latent-permutation method allows the model to learn an encoding explicitly in the latent space instead of implicitly in the encoder/decoder networks. This might be a powerful tool since it further constricts the bottleneck in the auto-encoder sense, although at the cost of suboptimally encoding identical latent spaces. This can be, and is, remedied by using both contrastive learning and latent-permutation in conjunction.

Having completely disentangled latent spaces is a local minimum in the latent-permutation method and allows for the ideal swap in Figure 2a. The CSLP-AE model is able to keep the latent spaces disentangled while also providing the structural encoding information required for the conversion method to work. This is evident from Table 1 which shows that the stand-alone latent-permutation model (SLP-AE) achieves about half ($\approx 51.7\%$) the MSE error of the C-AE model on the ERP conversion tasks using samples from different tasks or subjects (D.s, S.t. and S.s., D.t.), and the CSLP-AE model retains this performance. We propose the latent-permutation method as a replacement for the standard auto-encoder reconstruction loss to be used in conjunction with contrastive learning and the batch construction method to provide disentangled latent spaces which also allows for the structural encoding information to flow through and ease zero-shot conversion.

The latent-permutation method does not increase performance on the (D.s., D.t.) conversion task. Arguably the difference in performance between the stand-alone latent-permutation and contrastive learning methods (on D.s, S.t. and S.s., D.t.) is due to this structural encoding property. An analysis is provided in the appendix providing a generalization of the latent-permutation method onto different subject/different task conversion to circumvent the structural encoding reliability. Further research could explore different avenues for keeping structural information while also disentangling subject and task latents, or providing a source for this structural encoding using generative methods or distributional power to generate structure, e.g. through the use of a VAEs [30].

## 5   Conclusion

In this paper, a novel split-latent permutation framework was introduced for disentangling subject and task content in EEG signals and enabling single-trial zero-shot conversion. By combining the proposed split-latent permutation framework with contrastive learning, we achieved better performance compared to standard deep learning methods on the standardized ERP Core dataset. The experimental results demonstrated a significant 51.7% improvement in ERP conversion loss on unseen to unseen subject conversions. The method also achieved high single-trial subject classification accuracy (80.32±0.28%) and single-trial task classification accuracy (48.48±0.34%) on unseen subjects.

**Limitations**   The conversion results in this paper are limited to the single dataset on which they were trained, and may not generalize to other datasets with different experimental conditions. The ERP Core dataset is meant to standardize ERP measurements with more data to come in the future from other laboratories which might alleviate this limitation. Furthermore, the tasks used in the experiments are all seen during training. This limits the scope of the results to the seen tasks and no conclusions can be drawn about the generalization of the model to unseen tasks.

**Broader Impact**   Zero-shot conversion of EEG signals, similar to other deep fake methods, can be used for malicious purposes. Methods discussed in this paper intrude on one of the most sacred places yet to be exploited and confused by technology: the human mind. Malicious use of this technology includes the ability to decode thoughts and intentions from EEG signals, and the ability to create fake EEG signals to confuse verification systems using EEG signals as a biometric identifier [11, 37, 51]. Care must be taken when developing and deploying such technology to ensure that it is not used for malicious purposes. However, it can also be used to improve the quality of life for people with and without disabilities. It provides a base for generalizing EEG signal representations across subjects and tasks, which can be used to improve the performance of EEG-based BCI systems, especially on a single-trial level as attestable by the results in Table 1. Similarly, it could provide a base for novel analysis strategies, such as predicting drug or stimulus reactions in a healthy versus diseased brain, or as biomarkers of brain disorders.

## Acknowledgments and Disclosure of Funding

Funding in direct support of this work: Lundbeck Foundation grant R347-2020-2439.

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

# A    Appendix

## A.1    Structural encoding and generalization of split-latent permutation loss

We observed a tendency in the SLP-AE model trained using only the split-latent permutation loss, in which the model would simply learn identical latent spaces. We discussed how this tendency stems from the information-propagation through one of the latents during training. Split-latent permutation is trained using the self-reconstruction loss where one of the latents given as input to the decoder is swapped with that of another sample where both samples either belong to the same task or the same subject. Given that only one latent is swapped at a time, the model can learn a permutation-invariant encoding of the signal not specific to either the subject or task content of the signal, and it could rely on this information during (S.s., S.t.), (D.s., S.t.), (S.s., D.t.) conversion.

In this material we generalize the split-latent permutation using a quadruplet sampling method instead to instances where the input sample and the reconstruction target (output sample) is not the same, but which in special cases becomes identical to both the split-latent permutation and the self-reconstruction loss.

During training we sample a batch of $K$ quadruplets, $\{(X_k^a, X_k^b, X_k^c, X_k^d)\}_{k=1}^K$. For each index of the batch $k$ we choose two random subjects, $U_k$ and $V_k$, and two random tasks, $M_k$ and $N_k$. The quadruplet samples are sampled such that

$$X_k^a \text{ has subject and task } (U_k, M_k) \tag{6}$$

$$X_k^b \text{ has subject and task } (V_k, M_k) \tag{7}$$

$$X_k^c \text{ has subject and task } (U_k, N_k) \tag{8}$$

$$X_k^d \text{ has subject and task } (V_k, N_k) \tag{9}$$

Similar to the split-latent permutation, the encoding of these quadruplets should match and disentangle the subject and task content into their respective latent spaces, such that a latent-swap between two latents (which ideally encode the same information) has minimal impact on the reconstruction/conversion. With these quadruplets we can now generalize the split-latent permutation such that both latents are swapped with latents from other samples which should encode the same information. The samples encode the following latents

$$X_k^a \text{ encodes } (z_k^{(\mathcal{S},a)}, z_k^{(\mathcal{T},a)}) \tag{10}$$

$$X_k^b \text{ encodes } (z_k^{(\mathcal{S},b)}, z_k^{(\mathcal{T},b)}) \tag{11}$$

$$X_k^c \text{ encodes } (z_k^{(\mathcal{S},c)}, z_k^{(\mathcal{T},c)}) \tag{12}$$

$$X_k^d \text{ encodes } (z_k^{(\mathcal{S},d)}, z_k^{(\mathcal{T},d)}) \tag{13}$$

where

$$z_k^{(\mathcal{S},a)} \text{ and } z_k^{(\mathcal{S},c)} \text{ both ideally encode } U_k \tag{14}$$

$$z_k^{(\mathcal{S},b)} \text{ and } z_k^{(\mathcal{S},d)} \text{ both ideally encode } V_k \tag{15}$$

$$z_k^{(\mathcal{T},a)} \text{ and } z_k^{(\mathcal{T},b)} \text{ both ideally encode } M_k \tag{16}$$

$$z_k^{(\mathcal{T},c)} \text{ and } z_k^{(\mathcal{T},d)} \text{ both ideally encode } N_k \tag{17}$$

All of these pairs of latents which ideally encode the same information are swapped in the generalized split-latent permutation loss, which we will refer to as the *quadruplet permutation loss* (QP-loss). With this full swap of latents, there is no direct path between the input sample and the output sample, and the model is forced to encode the subject and task content into their respective latents. The reconstructions are as follows

$$\hat{X}_k^a = D_\phi(z_k^{(\mathcal{S},c)}, z_k^{(\mathcal{T},b)}) \text{ should reconstruct } X_k^a \tag{18}$$

$$\hat{X}_k^b = D_\phi(z_k^{(\mathcal{S},d)}, z_k^{(\mathcal{T},a)}) \text{ should reconstruct } X_k^b \tag{19}$$

$$\hat{X}_k^c = D_\phi(z_k^{(\mathcal{S},a)}, z_k^{(\mathcal{T},d)}) \text{ should reconstruct } X_k^c \tag{20}$$

$$\hat{X}_k^d = D_\phi(z_k^{(\mathcal{S},b)}, z_k^{(\mathcal{T},c)}) \text{ should reconstruct } X_k^d \tag{21}$$

**Figure 4:** Illustration of the quadruplet permutation loss. The quadruplet permutation loss is a generalization of the split-latent permutation loss, where the latents are swapped in pairs such that there is no direct path between the input sample and the reconstruction. The quadruplet permutation loss is illustrated with the quadruplet $(X_k^a, X_k^b, X_k^c, X_k^d)$ where the latents are swapped such that $\boldsymbol{z}_k^{(\mathcal{S},a)}$ and $\boldsymbol{z}_k^{(\mathcal{S},c)}$ are swapped, and $\boldsymbol{z}_k^{(\mathcal{T},a)}$ and $\boldsymbol{z}_k^{(\mathcal{T},b)}$ are swapped, etc., before decoding. This is done for all quadruplet samples yielding the quadruplet permutation loss as the MSE loss between the input sample and the reconstruction.

The swap of latents is illustrated in Figure 4.

The quadruplet permutation loss is defined as

$$\mathcal{L}_{\text{QP}} = \frac{1}{4K} \sum_{k=1}^{K} \left( \|X_k^a - \hat{X}_k^a\|_2^2 + \|X_k^b - \hat{X}_k^b\|_2^2 + \|X_k^c - \hat{X}_k^c\|_2^2 + \|X_k^d - \hat{X}_k^d\|_2^2 \right) \qquad (22)$$

The quadruplet permutation loss collapses to the split-latent permutation loss in some special cases. When input samples $X_k^a$ and $X_k^c$ are the same, then it becomes the same-subject permutation loss, and when input samples $X_k^a$ and $X_k^b$ are the same, then it becomes the same-task permutation loss. In the case where all input samples are the same, then the quadruplet permutation loss becomes the self-reconstruction loss.

**Quadruplet permutation loss results**

We provide here results using the same training and testing setup as in the paper. We conduct an ablation study on contrastive learning, latent-permutation and quadruplet permutation loss. The following four models are trained in five repetitions each:

- **SQP-AE**: Quadruplet permutation loss only.
- **CSQP-AE**: Quadruplet permutation loss and contrastive loss in conjunction.
- **SQLP-AE**: Quadruplet permutation loss and latent-permutation loss in conjunction.
- **CSQLP-AE**: Quadruplet permutation loss, contrastive loss and latent-permutation loss in conjunction.

Here we see that the quadruplet permutation loss degrades performance on subject classification accuracy considerably, but with similar performance on task classification accuracy. We also see that

**Table 3:** Single-trial balanced subject classification accuracy (S.acc%), task-on-subject classification accuracy (T⊢S.acc%), task classification accuracy (T.acc%), subject-on-task classification accuracy (S⊢T.acc%), and zero-shot same-subject same-task ERP conversion MSE (S.s., S.t.), different-subject different-task ERP conversion MSE (D.s, D.t.), different-subject same-task ERP conversion MSE (D.s., S.t.), same-subject different-task ERP conversion MSE (S.s., D.t.). All ERP conversion MSE values have scales of $10^{-11}V^2$. Epoch window was 1s.

| Model | S.acc% | T⊢S.acc% | T.acc% | S⊢T.acc% | (S.s., S.t.) | (D.s, D.t.) | (D.s., S.t.) | (S.s., D.t.) |
|---|---|---|---|---|---|---|---|---|
| CSQLP-AE | $76.10 \pm 0.76$ | $43.36 \pm 0.46$ | $46.17 \pm 0.25$ | $76.47 \pm 0.46$ | $1.91 \pm 0.08$ | $6.90 \pm 0.05$ | $3.43 \pm 0.05$ | $3.94 \pm 0.16$ |
| SQLP-AE | $69.26 \pm 0.50$ | $46.86 \pm 1.35$ | $44.80 \pm 0.77$ | $69.60 \pm 0.25$ | $\mathbf{1.48 \pm 0.05}$ | $\mathbf{6.44 \pm 0.03}$ | $\mathbf{2.87 \pm 0.10}$ | $\mathbf{2.97 \pm 0.05}$ |
| CSQP-AE | $73.04 \pm 0.50$ | $35.89 \pm 0.42$ | $44.83 \pm 0.21$ | $71.56 \pm 0.64$ | $6.20 \pm 0.12$ | $7.00 \pm 0.08$ | $6.60 \pm 0.11$ | $6.59 \pm 0.10$ |
| SQP-AE | $73.44 \pm 0.33$ | $43.92 \pm 0.29$ | $\mathbf{48.88 \pm 0.13}$ | $70.42 \pm 0.39$ | $5.58 \pm 0.07$ | $6.49 \pm 0.04$ | $6.07 \pm 0.04$ | $5.99 \pm 0.06$ |
| CSLP-AE | $\mathbf{80.32 \pm 0.28}$ | $45.41 \pm 0.37$ | $48.48 \pm 0.34$ | $79.64 \pm 0.37$ | $4.21 \pm 0.12$ | $20.06 \pm 0.10$ | $5.80 \pm 0.15$ | $6.65 \pm 0.23$ |
| SLP-AE | $74.63 \pm 0.74$ | $47.23 \pm 0.31$ | $47.00 \pm 0.32$ | $74.70 \pm 0.73$ | $3.82 \pm 0.04$ | $19.92 \pm 0.10$ | $6.12 \pm 0.09$ | $5.02 \pm 0.08$ |
| C-AE | $79.42 \pm 0.48$ | $37.34 \pm 0.45$ | $46.59 \pm 0.23$ | $73.27 \pm 0.25$ | $4.28 \pm 0.06$ | $20.28 \pm 0.07$ | $11.33 \pm 0.47$ | $10.64 \pm 0.30$ |

the (D.s., D.t.) conversion loss now is on the same level as (D.s., S.t.) and (S.s., D.t.) conversion for the models without latent permutation. Adding the latent-permutation, although it introduces the structural encoding pathway to the model again, considerably decreases both (S.s., S.t.) conversion compared to CSLP-AE and SLP-AE.

We provide $t$-SNE [1, 31, 65] plots of the generalized models here and the CSLP-AE model in Figure 5a. Furthermore, we provide (D.s., D.t.) conversion examples in Figure 5b for the generalized models and the CSLP-AE model.

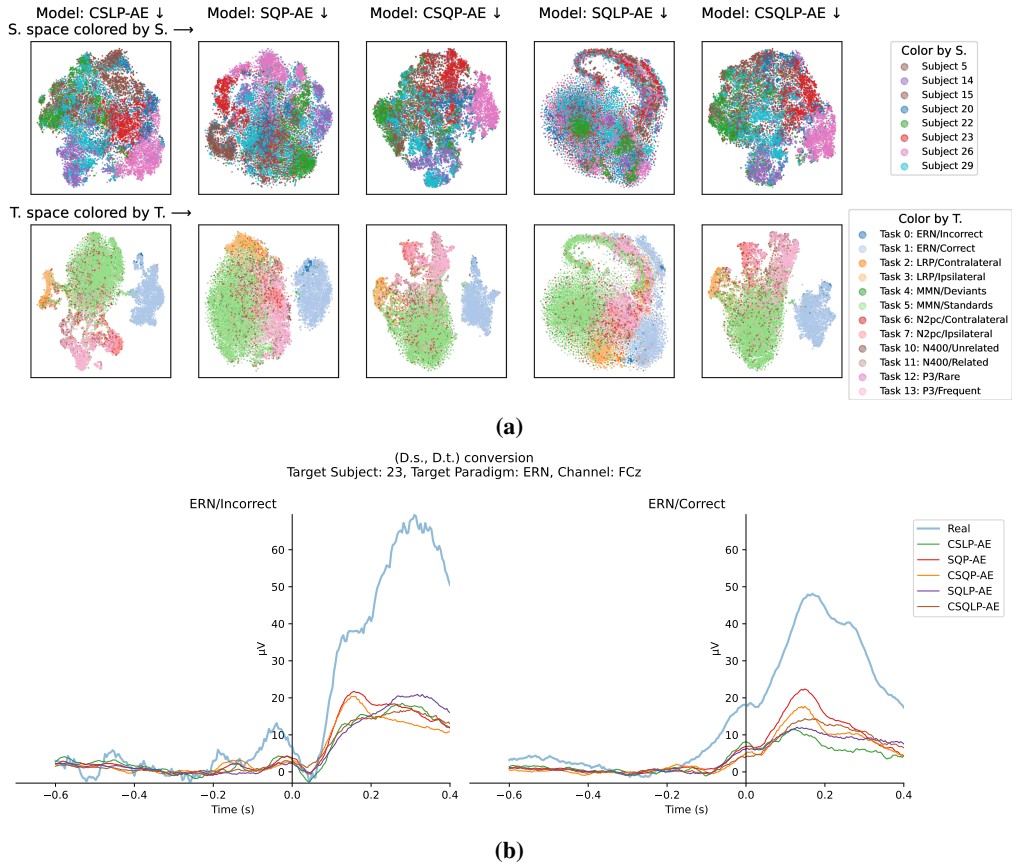

**Figure 5: (a)** $t$-SNE plots of split-latents as encoded on the test set (unseen subjects), colored by true labels. Rows show *subject* and *task* latent spaces, respectively, while columns indicate models CSLP-AE, SQP-AE, CSQP-AE, SQLP-AE, and CSQLP-AE respectively. **(b)** (D.s., D.t.) converted ERPs from the same five models for a random target subject and target paradigm. All latents used in conversion were from unseen subjects on the test set, i.e. unseen to unseen conversion.

The SQP-AE model achieves specialized latent spaces with disentangled subject and task content as evident from the $t$-SNE plots in Figure 5a. Notably, the latent space is similar to the CSLP-AE model, but using simply the quadruplet permutation loss. In this sense, the quadruplet permutation loss is similar to the contrastive loss in that it encourages the model to learn a disentangled latent space while also directly learning all conversion schemes. Further research could focus on this quadruplet permutation loss and its relation to the contrastive loss. We view it as a contrastive loss that relates input samples to the output sample (the reconstruction), i.e. an auto-encoder contrastive loss, whereas a standard contrastive loss operates in the latent space itself. When we add the latent-permutation loss back to the model, we see that the structural encoding property occurs again in the SQLP-AE model, while the CSQLP-AE model does not have this property due to the contrastive loss used in specializing the latent space. Therefore, the SQP-AE model might be most comparable to the CSLP-AE model from the paper, as it both optimizes for conversion and latent disentanglement.

## A.2 Loss component scaling

In this section we provide loss progressions on the ERP Core dataset for 200 epochs with standard error of the mean over the different runs shown as confidence bounds.

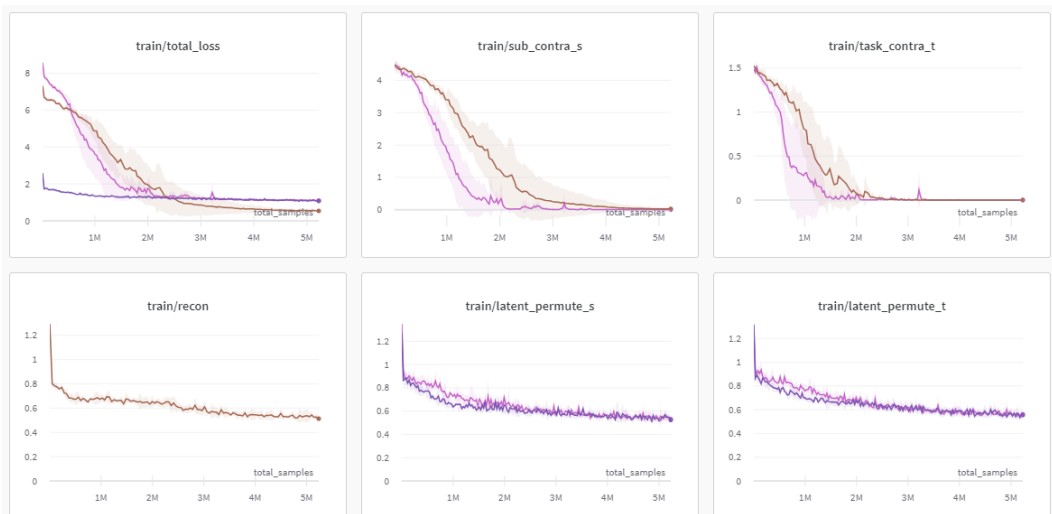

**Figure 6:** Training loss progressions on ERPCore [iv] dataset. Total loss is sum of loss components. *sub_contra_s* and *task_contra_t* are contrastive losses in the subject and task latent spaces respectively. *recon* is the reconstruction loss. *latent_permute_s* and *latent_permute_t* are the latent permutation loss terms respectively. The pink loss progressions are from CSLP-AE, the purple progressions are from SLP and the orange/brown progressions are from C-AE.

## A.3 Glossary of symbols and abbreviations

**Table 4:** Comprehensive glossary over symbols and abbreviations

| Symbol/Abbreviation | Meaning |
|---|---|
| EEG | Electroencephalography |
| ERP | Event-related potential |
| BCI | Brain-computer interfacing |
| CV | Cross-validation |
| MSE | Mean squared error |
| AE | Auto-encoder |
| CNN | Convolutional Neural Network |
| FHVAE | Factorized Hierarchical Variational Autoencoder |
| CSP | Common Spatial Pattern |
| CSLP-AE | Contrastive Split-Latent Permutation Autoencoder |
| SLP-AE | Split-Latent Permutation Autoencoder |
| C-AE | Contrastive Autoencoder |
| CL | Encoder using contrastive loss |
| CE | Supervised Encoder using cross-entropy |
| CE(t) | Supervised Encoder using cross-entropy in task space only |
| $E_\theta(\cdot)$ | Encoder parameterized by $\theta$ |
| $D_\phi(\cdot)$ | Decoder parameterized by $\phi$ |
| $\boldsymbol{X}$ | EEG Sample used as input to encoder |
| $\mathcal{S}$ | Subject latent space |
| $\mathcal{T}$ | Task latent space |
| $\mathcal{L}$ | A given latent space (one of subject or task) |
| $\boldsymbol{z}^{(\mathcal{S})}$ | Subject latent |
| $\boldsymbol{z}^{(\mathcal{T})}$ | Task latent |
| $\hat{\boldsymbol{X}}$ | Reconstructed EEG sample from output of decoder |
| $(\boldsymbol{X}_i^a, \boldsymbol{X}_i^b)$ | Pair of EEG samples at $i$'th batch index with either a common subject or task |
| $(\boldsymbol{z}_i^{(\mathcal{S},a)}, \boldsymbol{z}_i^{(\mathcal{T},a)})$ | Subject and task latent from encoding $\boldsymbol{X}_i^a$ |
| $(\boldsymbol{z}_i^{(\mathcal{S},b)}, \boldsymbol{z}_i^{(\mathcal{T},b)})$ | Subject and task latent from encoding $\boldsymbol{X}_i^b$ |
| $\hat{\boldsymbol{X}}_i^{(\mathcal{T},a)}$ | Reconstructed EEG sample of $\boldsymbol{X}_i^a$ where task latents are swapped in latent space $\mathcal{T}$ |
| $\hat{\boldsymbol{X}}_i^{(\mathcal{T},b)}$ | Reconstructed EEG sample of $\boldsymbol{X}_i^b$ where task latents are swapped in latent space $\mathcal{T}$ |
| $L_{LP}(\mathcal{L}; \boldsymbol{X}_i^a, \boldsymbol{X}_i^b)$ | Split-latent permutation loss which swaps latents in latent space $\mathcal{L}$ and where $\boldsymbol{X}_i^a$ and $\boldsymbol{X}_i^b$ have either common subject or task corresponding to $\mathcal{L}$ |
| $L_{\text{NT-Xent}}(\mathcal{L}; \cdot, \cdot, k)$ | $N$-pair cross-entropy loss in latent space $\mathcal{L}$ for the $k$'th row |
| $L_{\text{CLIP}}(\mathcal{L}; \cdot, \cdot)$ | Symmetric temperature-scaled cross-entropy loss in latent space $\mathcal{L}$ |
| (S.s., S.t.) | Same subject, same task conversion scheme |
| (S.s., D.t.) | Same subject, different task conversion scheme |
| (D.s., S.t.) | Different subject, same task conversion scheme |
| (D.s., D.t.) | Different subejct, different task conversion scheme |
| $\sigma$ | Target subject |
| $\gamma$ | Target task |
| $\hat{x}_k^{(\sigma,\gamma)}$ | $k$'th reconstructed EEG from targets |
| $\hat{x}_{(\sigma,\gamma)}^{\text{C-ERP}}$ | Average ERP from conversion with sampled targets |
| $\hat{x}_{(\sigma,\gamma)}^{\text{ERP}}$ | Average ERP from ground truth targets |

