# OpenReview forum: "CSLP-AE: A Contrastive Split-Latent Permutation Autoencoder Framework for Zero-Shot Electroencephalography Signal Conversion"
_NeurIPS.cc/2023/Conference — NeurIPS 2023 poster_

### Official Review · Reviewer_MXTP · 2023-06-25

**Soundness:** 2 fair
**Presentation:** 3 good
**Contribution:** 2 fair
**Rating:** 5
**Confidence:** 4

**Summary:**

This research paper introduces a novel framework called the contrastive split-latent permutation autoencoder (CSLP-AE) for EEG signal conversion, with the primary aim of disentangling task-related content and subject-specific style. This technique is based on disentangled representation learning which has been widely used in CV domains such as gait analysis and face recognition.

The authors hypothesize that the conversion of EEG signals between tasks and subjects necessitates the learning of latent representations that account for both content and style. They employ contrastive learning to guide these latent splits to represent the subject (style) and task (content) separately.

The efficacy of the CSLP-AE framework was tested on a standardized ERP dataset, ERP Core, which contains data from the same subjects across various standardized paradigms. The authors contrasted the performance of the CSLP-AE against conventional supervised, unsupervised, and self-supervised training.

The results of the study showed that the proposed framework significantly improved performance compared to standard deep learning methods on unseen-to-unseen subject conversions by 51.7%. Furthermore, it achieved high accuracy in single-trial subject classification (80.32±0.28%) and single-trial task classification (48.48±0.34%) on unseen subjects.

In essence, this study provides a novel and effective method for EEG signal conversion, significantly enhancing the accuracy and interpretability of EEG analyses. It further paves the way for potential applications in a broader range of biological signal-processing tasks.

**Strengths:**

1. The introduction of the contrastive split-latent permutation autoencoder (CSLP-AE) for EEG ERP classification is an innovative contribution to the field. By adopting disentangled representation learning from other domains to the EEG domain, the work paves the way for new avenues of research and applications in the bio-signal domain.

2. The figures representing the framework architecture and different training tasks are very clear. This visual clarity significantly enhances the reader's understanding of the proposed method and its applications.

3. The methods section contains detailed and coherent explanations, which enables an effective understanding of the process and intricacies involved in the proposed model.

4. The manuscript provides a comprehensive comparative analysis of the proposed method with/without some loss components which verified the contribution of each proposed task. The results demonstrated the necessity and impact of each component, thereby fortifying the empirical robustness of the work.



**Weaknesses:**

1. While the diagrams and colours used in the Methodology section are commendable, the notations used in the methods section could be improved for enhanced comprehension. Providing a clear and comprehensive key for symbols and abbreviations would be beneficial for readers.

2. The manuscript could further elaborate on how the model accommodates noise caused by other factors within the EEG data. Clarification on whether this noise is modelled within subject-specific or task-related features would be beneficial. Additional analysis and discussion on this topic would bolster the paper's validity.

3.  Table 1 indicates effective feature disentanglement as evidenced by the comparison between T-on-S and S-on-T. However, the difference between S and S-T within the proposed model is not significantly different, suggesting subject-specific information is still greatly present within the T latent features. It would be beneficial to discuss potential reasons for this observation and how it aligns with the proposed model's disentanglement claim.

4. The textual content in Figure 3 is slightly difficult to read due to its small size. Additionally, presenting t-SNE separately for S and T spaces may not intuitively display feature disentanglement. It might be more effective to reduce the total number of points for each colour and integrate both spaces into a single image for each model, using colour to represent S and shape variations to represent T. This modification could enhance the visual clarity and intuitiveness of the model's results.

5. The paper currently offers limited evaluation of the reconstructed signal quality, with a single plot in the time domain (Figure 3. B). It would be valuable to include evaluation in the frequency domain as well. Implementing additional quantitative metrics such as the Inception Score and Frechet Inception Distance could supplement the existing Euclidean distance measure and thus, provide a more comprehensive evaluation of the reconstruction quality.

6. While the paper presents compelling results, it could be further strengthened by providing more extensive comparisons with current state-of-the-art methods. For the classification tasks, a comparison with sophisticated EEG decoding methods based on CNNs would be beneficial. Similarly, for the reconstruction tasks, contrasting the proposed model's performance with other well-established techniques based on AE, VAEs and GANs would be insightful. These comparisons would help situate the proposed method within the larger context of cutting-edge techniques and provide a more robust validation of its performance.

7. For improved reproducibility, it would be beneficial to provide a detailed outline of the model architecture and hyperparameters. For instance, the weight allocation between each loss component would be incredibly important for researchers seeking to reproduce or extend this work.





**Questions:**

1. Could the authors specify whether any part of the network was frozen during different training tasks? Providing this information would help in understanding the training dynamics of the proposed model.

2. In the baseline comparisons, the authors referred to 'CE' models trained with supervised learning cross-entropy loss. Could the authors detail the structure of the model used in this supervised training process? It would be beneficial if the authors could use well-established EEG decoding models as classification baseline comparisons. This would facilitate comparison with previously published results and provide more context for the performance of the proposed model.


**Limitations:**

Please refer to weaknesses

---

> ### Author Rebuttal · Authors · 2023-08-10
>
> We thank the reviewer for the careful and constructive critique acknowledging that the proposed approach is an innovative contribution to the field with visual clarity, detailed and coherent explanations, and a comprehensive comparative analysis of the proposed method.
> ### Re. Clear and comprehensive key for symbols and abbreviations.
> We appreciate the comment and will include such comprehensive information in the revised manuscript.
> ### Re. Accommodation of noise caused by other factors within the EEG data.
> This noise is discussed in the manuscript as part of the structural encoding discussion, see line 305. Specifically, we consider structural information also constituted by noise. The AE objective will focus on also recovering noise-structures in the reconstruction admitted by the encoding of structural information. In the supplementary material, we discuss how the structural information can be blocked from the decoder by considering a quadruplet latent permutation (QLP, see Supplementary Figure 1, section A). Notably, QLP is also found to explicitly enhance conversion between different tasks and different subjects (D.s, D.t.) (see Supplementary Table 1).
> ### Re. Subject-specific information is still greatly present within the T latent features.
> We agree that subject information is still present in the task latent space and vice versa for the subject latent space, and we attribute this to the ability of the decoder to extract subject and task information from the respective spaces in which the two latent representations can be identically encoded (see also the discussion of this and the motivation for the use of contrastive learning in combination with the split latent permutation framework in lines 144-152). We further attribute this to inclusion of structural information and demonstrate how this can be remedied considering the QLP explored in section A of the supplementary material. Whereas contrastive learning promotes latent spaces to characterize task and subjects respectively as shown in Table 1, subject and task information is still preserved in the opposing latent spaces. We attribute this to the contrastive loss successfully promoting subject and task disentanglements in respective spaces; however, the high capacity of the spaces still accommodates a large degree of subject and task information. Careful bottleneck tuning can remedy this, for a discussion in the context of voice conversion see [40].
> ### Re. Textual content in Figure 3 is slightly difficult to read due to its small size.
> We apologize for the lack of clarity on the textual content of Figure 3, we will improve this in the revised manuscript. Less points could be a possibility, but the current t-SNE shows the topological similarities between different model latent spaces as intended. t-SNE plots showing the task latent space colored by subject and the subject latent space colored by task are provided in the supplementary material Section F.3.
> ### Re. The paper currently offers limited evaluation of the reconstructed signal quality.
> Due to space limitations the main paper only included a single conversion plot in Figure 3.B. However, in the supplementary material section H we included further conversion plots on both test and training data for all paradigms and conversion schemes.
> We quantitatively evaluate conversion reconstruction performance using the mean-squared-error metric as this is what the AE objective function is minimizing, and the mean square error loss is also optimal regarding minimizing mean discrepancies. Finally, the evoked response is calculated as the average across trials which is also least squares optimal. We therefore considered the MSE metric as opposed to  other procedures such as the inception score, Frechet inception distance, the scale-invariant signal to distortion, or correlation metric typically used in image and audio-processing.
> ### Re. More extensive comparisons with current state-of-the-art methods.
> We agree that additional analyses and comparison to current SOTA task classification will improve the manuscript and have therefore included two additional datasets on Motor Imagery (MI) [x] and sleep stage classification [v] as part of the Author Rebuttal and included a comparison to SOTA deep learning procedures for task classification on these datasets. We find find that our approach provides on par performance for MI and can even operate in the situation where data is not time-locked (I.e., sleep stage classification) although not at the level of current SOTA (see also the included Author Rebuttal associated PDF).
> ### Re. For improved reproducibility, it would be beneficial to provide a detailed outline of the model architecture and hyperparameters.
> This is included in both the manuscript and the supplementary materials. See line 210-214 in the manuscript as well as Supplementary section C and D.
> ### Re. Could the authors specify whether any part of the network was frozen during different training tasks?
> No parts of the network were frozen during different training tasks. Since losses were found to on similar scales (see Author Rebuttal PDF), we chose to use a complete loss function rather than stepwise minimizing the latent space losses individually.
> ### Re. Could the authors detail the structure of the CE model used in this supervised training process?
> The CE models with supervised learning use the same encoder as the other models, i.e., a strided CNN and a transformer bottleneck. This is very much akin to the t-CTrans model from [xii] although with strided convolutions instead of average pooling.
> ### Re. Well-established EEG decoding models as classification baseline comparisons
> We have for the two new included datasets also included comparison to current SOTA  for these datasets (see results in Author Rebuttal and PDF) in which we find that our procedure provides on par performance to current prominent SOTA methods for MI.
> ### References: See Author Rebuttal.

---

> > ### Comment · Reviewer_MXTP · 2023-08-12
> >
> > Thanks for the authors' responses and efforts. I have upgraded the Rating to Borderline accept

---

### Official Review · Reviewer_4KfP · 2023-07-01

**Soundness:** 3 good
**Presentation:** 3 good
**Contribution:** 3 good
**Rating:** 6
**Confidence:** 4

**Summary:**

Electroencephalography (EEG), a non-invasive neuroimaging technique, is often compromised by high noise levels and subject variability, making consistent signal extraction challenging. To mitigate this, the authors introduce a novel framework, the Contrastive Split-Latent Permutation Autoencoder (CSLP-AE). This framework enhances EEG conversion using contrastive learning, guiding latent representations and enabling the clear distinction of subject variability (style) and task activation (content). Beyond providing a better generalizable understanding of subjects and tasks, this technique also facilitates conversion between unseen subjects. This suggests it has a wider potential for signal conversion, content extraction, and style extraction in biological signal analysis.

**Strengths:**

The paper is innovative in being among the first to separate style and content in EEG signals.

The proposed method outperforms its competitors in experimental tests.

**Weaknesses:**

While the concept of disentangling content from style has been extensively discussed in the literature, the authors did not compare their method with these existing approaches in their experiments.

In Equation (4), it seems that $L_{NT-Xent}(L; Z', Z'', k) = L_{NT-Xent}(L; Z'', Z', k)$ if we choose the cosine similarity as the similarity metric. The necessity for Equation (5) is not fully clear. The authors also do not explain how to integrate Equation (3) with Equation (5) in the optimization process.

The declared goal of the proposed method is to divide the representations into subject and tasks. However, the authors did not provide concrete evidence demonstrating the achievement of this objective. Is it that the subject representations can predict the subject but not the task, and vice versa?

**Questions:**

Could you provide a comparison of your method with other established approaches that disentangle content from style in the literature?

With regards to Equation (4), it appears that if the cosine similarity is chosen as the similarity metric,  $L_{NT-Xent}(L; Z', Z'', k) = L_{NT-Xent}(L; Z'', Z', k)$. Could you clarify why Equation (5) is then necessary?

The integration of Equation (3) with Equation (5) in the optimization process is not explicitly explained. Could you provide more details on how these equations are combined in the optimization?

The declared goal of the proposed method is to divide the representations into subject and tasks. How have you confirmed that this objective is met? Specifically, can subject representations predict the subject but not the task, and vice versa?

**Limitations:**

The authors addressed the limitations and potential negative social impact of their work.

---

> ### Author Rebuttal · Authors · 2023-08-10
>
> We thank the reviewer for the careful assessment of the manuscript acknowledging that the paper is innovative in being among the first to separate style and content in EEG signals providing favourable performance and that it may have potential for wider adoption within biomedical signal processing and analysis problems.
> ### Re. Could you provide a comparison of your method with other established approaches that disentangle content from style in the literature?
> As also remarked to reviewer VVQF, we did not include comparisons to the existing EEG literature [4,37,42] for a few different reasons. In [4], the task classification setup is based on binary classification based on samples of same vs. different task as opposed to the multi-class task classification problem presently considered in the manuscript.
> [37] and [42] did not produce explicit subject latent representations but used adversarial training to achieve latent representations that are subject invariant. This is a slightly different aim than ours, which is to explicitly provide both subject and tasks latent representations. We agree that further comparison to state-of-the-art would strengthen the paper, and we have therefore included results on two additional datasets on movement/imagery [x, iii] and sleep stage classification [v, iii] using the same framework and model architecture as presented in the original manuscript. In the supplied Author Rebuttal PDF, we contrast our performance to current state-of-the-art on these two benchmark datasets. We observe on-par performance of our proposed approach to current SOTA baselines for the MI whereas the model can also operate for sleep-stage classification, i.e., on data that is not time-locked although at performance inferior to SOTA sleep stage classification procedures.
> ### Re. Clarification of necessity of Equation (5)
> Whereas Equation (4) considers a given sample $k$ to all other samples, Equation (5) considers all combinations of pairs by summing over $k$. By doing so with both ($Z^A$, $Z^B$) and ($Z^B$, $Z^A$), the contrast is ensured to be based on all pair-wise combinations, as the normalization in Equation (4) is summing across the second latent embedding matrix where $i≠k$. The necessity of equation (5) is simply to provide a symmetrical contrastive loss across both rows and columns of the corresponding similarity matrix. This is also the loss used in [41].
> ### Re. Integration of Equation (3) with Equation (5) in the optimization process.
> We acknowledge that this should have been made clearer in the manuscript, and the revised version will reflect this. The two loss components Equation (3) and Equation (5) are combined into a joint loss function by adding the two loss components equally, i.e., Equation (3) and Equation (5) are both weighted by one in the optimization process, such that CSLP-AE minimises $L =L_{\mathrm{LP}}(\mathcal{S}; ·, ·)+ L_{\mathrm{LP}} (\mathcal{T}; ·, ·)+ L_{\mathrm{CLIP}}(\mathcal{S}; ·, ·) + L_{\mathrm{CLIP}}(\mathcal{T} ; ·, ·)$ (see also the description around line 194 of the manuscript). This means that Equation (5) is calculated for both the subject and task latents and added to the loss. Since all loss components have similar scale (as shown in the Author Rebuttal PDF) weighing each term the same (by one) was found to be justified.
> ### Re. The declared goal of the proposed method is to divide the representations into subject and tasks. How have you confirmed that this objective is met? Specifically, can subject representations predict the subject but not the task, and vice versa?
> We use classification performance to quantitatively evaluate the representations in characterizing subject and tasks as shown in Table 1. Specifically, we show the classification performances for task-on-subject (T⊢S) and subject-on-task (S⊢T) problems. Here, T⊢S refers to classifying tasks based on the subject latents, and vice-versa for the S⊢T problem. This is also detailed in lines 222-227 in the manuscript with supporting information concerning classifier training details in Section D.2 in the supplementary material. We would also like to refer to the discussion concerning structural encoding as an explanation for the (seemingly) similar latent space topologies (see lines 274-335 in the manuscript).
> ### References: See Author Rebuttal.

---

> > ### Comment · Reviewer_4KfP · 2023-08-16
> > **Comparing the DREAM Model**
> >
> > I recently came across the DREAM model presented in [1]. Unlike the proposed method, the DREAM model employs self-supervised learning to disentangle subject-related and task-related representations, instead of using a permutation strategy. Could you elucidate the advantages of the proposed method in comparison to the DREAM model? I am sorry that I did not include this question in my review earlier, as I read the paper after the review submission.
> >
> > References:
> > [1] Lee, Seungyeon, Thai-Hoang Pham, and Ping Zhang. "DREAM: Domain Invariant and Contrastive Representation for Sleep Dynamics." In 2022 IEEE International Conference on Data Mining (ICDM), pp. 1029-1034. IEEE, 2022.

---

> > > ### Author Response · Authors · 2023-08-17
> > > **The merits of CSLP-AE as opposed to DREAM**
> > >
> > > As pointed out by the reviewer, DREAM is in the family of contrastive autoencoder (AE) approaches (C-AE) for EEG. Similar to [4], DREAM uses a *variational* AE as opposed to AE. Both DREAM, [4] and the proposed method explicitly operate in two latent spaces encouraged to represent subject and task respectively by contrastive learning. Whereas [4] uses a classification loss for the contrast, DREAM uses both similarity-based contrastive learning (as our C-AE and CSLP-AE) and supervised classification losses imposed on each of the two latent spaces. The advantages of the CSLP-AE in comparison to DREAM are thus the advantages of the split-latent permutation training as opposed to conventional C-AE to which we systematically compare against. Namely, that our latent permutation procedure explicitly promotes the latent representations to account for subject and task in the reconstruction loss by the imposed permutation. Notably, we find that this enhances performance when compared to conventional C-AE. Furthermore, our CSLP-AE provides a complementary framework to promote the extraction of generalizable patterns of subject and task content that is generic, and which can be added to existing C-AE procedures including the variational inference procedure used in [4] and DREAM. Importantly, the split-latent permutation framework further enables EEG signal conversion and to probe what the response would be, i.e., had a given subject performed a different task, or a given task been performed by a different subject.

---

> > > > ### Comment · Reviewer_VVQF · 2023-08-17
> > > > **missing ablation for sleep staging in rebuttal PDF**
> > > >
> > > > I concur with the authors that the split-latent permutation reconstruction cost has the potential to improve performance (and can be used for counterfactual reconstruction) as shown with the ERP and MI data. However, DREAM is designed and applied to sleep staging, where the split-latent permutation reconstruction cost may not be beneficial, but clearly the contrastive learning is helpful. Note that DREAM's performance is 83.91±5.62 on SleepEDF-20.
> > > >
> > > > I note that in the rebuttal PDF, there are no ablation tests (C-AE and SLP-AE) reported as were reported on ERP and MI. Can the authors report the results of the ablation on sleep data?  I'm generally skeptical that SLP helps in cases with less time-locking. This doesn't invalidate the contribution it simply clarifies the scope where the method is expected to help beyond split latent contrastive learning.

---

> > > > > ### Author Response · Authors · 2023-08-18
> > > > > **The results of C-AE and SLP-AE on Sleep-EDF Expanded [v, iii]**
> > > > >
> > > > > We thank the reviewer for following up on our response. As requested by the reviewer, we have now run the rest of the ablated models on the Sleep-EDF Expanded [v, iii] dataset using the same hyperparameters and run-time (50 epochs) as the earlier run CSLP-AE model, i.e., no tuning. The results are as follows:
> > > > >
> > > > > | Model              | Task Accuracy (%) |
> > > > > |--------------------|-------------------|
> > > > > | CSLP-AE            | 75.16 ± 0.95      |
> > > > > | SLP-AE             | 70.59 ± 1.18      |
> > > > > | C-AE               | 75.16 ± 0.86      |
> > > > >
> > > > > We see from the ablations that the CSLP-AE and C-AE indeed provide very similar performance.  Consequently, the use of split-latent permutation instead of conventional reconstruction loss when including contrastive learning does not seem to influence performance. However, contrastive learning is important here and as the reviewer points out the SLP-AE does not help the task-classification in this non-time-locked domain. We will include a discussion of this point along with the above results in the revised manuscript.

---

### Official Review · Reviewer_qodj · 2023-07-04

**Soundness:** 2 fair
**Presentation:** 2 fair
**Contribution:** 2 fair
**Rating:** 6
**Confidence:** 3

**Summary:**

The authors hypothesize that converting EEG signals between tasks and subjects requires extracting latent representations that capture both content and style. They introduce a novel framework called CSLP-AE, which utilizes contrastive learning to guide the latent representations and achieve EEG conversion. The proposed approach outperforms conventional training methods, providing reliable characterizations of subject and task, and allowing for zero-shot conversion of unseen subjects. The CSLP-AE framework has the potential for broader applications in signal conversion and the analysis of biological signals.


**Strengths:**

The paper uniquely frames a voice conversion problem as an EEG analysis problem. The paper presents a new framework that leverages state-of-the-art deep learning models to perform this disentanglement task. The paper clearly describes the benefit of disentangling the content and style of an EEG signal. The paper also clearly describes the model architecture and the training objective.

A main strength is the utilization of contrastive learning to guide the latent splits, effectively representing both subject (style) and task (content) aspects. By incorporating contrastive learning into the framework, the model gains the ability to disentangle and explicitly capture the unique characteristics associated with subjects and tasks, enhancing the overall interpretability and performance of the model.

**Weaknesses:**

The main weakness of the paper is that it does show results for more datasets. The task- and subject- latent spaces were not quantitively compared to see how cluster separation improved with the CSLP-AE.

Limited generalizability: The findings are constrained to the single dataset utilized in the experiments, despite efforts made to standardize protocols.

Limited task representation: The tasks employed in the experiments were exclusively those seen during training. Consequently, the scope of the results is confined to these known tasks, and the ability to draw conclusions regarding the model's generalization to unseen tasks is limited.

Restriction to time-locked paradigm: Only epochs with stimuli were included in the analysis, which restricts the scope of the conversion and representations to the time-locked paradigm. This limitation overlooks the potential application of the model to continuous cases, hindering a comprehensive analysis.

Lack of dataset diversity: The reliance on a single dataset, although standardized, may limit the representation of diverse experimental conditions. The inclusion of data from other laboratories in the future, as anticipated with the ERP Core dataset, could help mitigate this limitation.

**Questions:**

Why is the attention mechanism is necessary in this model? Why in Figure 3 does the task latent space t-SNE qualitatively show better separation of the tasks than the subject t-SNE if the CSLP-AE performed better on the subject classification task?

Figure 3A reveals that SLP-AE exhibits a topologically similar embedding, whereas C-AE and CSLP-AE do not demonstrate a similar pattern. It would be beneficial to provide additional explanation for this discrepancy. Additionally, the t-SNE embeddings of T. space from C-AE and CSLP-AE show similarity, despite the author's claim that Split-Latent Permutation is a key contribution. Further clarification on this aspect would be helpful.

**Limitations:**

The generalizability of the model to different datasets and tasks is unknown. While the author acknowledges the limitation of relying on a single dataset, it would be beneficial to explore the applicability of their findings by conducting tests on other datasets with different experimental conditions. This would provide a more comprehensive understanding of the generalizability of the proposed approach.

---

> ### Author Rebuttal · Authors · 2023-08-10
>
> We thank the reviewer for the careful review and constructive comments and for acknowledging that the paper uniquely frames a voice conversion problem as an EEG analysis problem presenting a new framework that leverages state-of-the-art deep learning models.
> ### Re. Limited generalizability and lack of dataset diversity.
> We will in the revised paper include two additional datasets considering tasks not covered by ERPCore, namely, the Movement/Imagery dataset [x] involving MI classification and the Sleep EDF Expanded sleep stage classification dataset [v] as detailed in the Author Rebuttal.
> ### Re. Restriction to time-locked paradigm.
> To address this limitation, we consider the sleep-stage classification task as outlined in the Author Rebuttal, which is not time-locked. We here observe that the procedure is able to characterize sleep-stages and thus not restricted to the time-locked paradigm, however, whereas it provides on par performance with SOTA for MI we find it to be inferior to current SOTA for sleep stage classification. We argue that this could be remedied by using encoders with larger receptive fields such as WaveNet [xv] instead of strided convolutions, which was also mentioned in the manuscript. For results and comparison to current state-of-the-art on these two additional datasets please see the included Author Rebuttal PDF.
> ### Re. The task- and subject- latent spaces were not quantitively compared to see how cluster separation improved with the CSLP-AE.
> We presently quantitatively evaluate the subject and task latent spaces by their ability to discriminate tasks and subjects which is commonly used to evaluate representation learning procedures (see for instance [xiv]. We agree that unsupervised clustering approaches could also be used for quantitative evaluation of cluster separation; however, this will involve several additional design choices, which will make comparative analysis difficult.
> ### Re. Why is the attention mechanism necessary in this model?
> Transformer architectures have proven to provide state-of-the-art results across domains. In particular, their ability to efficiently learn and leverage contexts when extracting latent representations have been widely explored. We therefore included attention mechanisms through the Transformer architecture to leverage this capability.  Transformers have previously successfully been applied in the context of EEG data in [ii], [ix], and [xii] which we use as comparison in the Author Rebuttal.
> ### Re. Why in Figure 3 does the task latent space t-SNE qualitatively show better separation of the tasks than the subject t-SNE if the CSLP-AE performed better on the subject classification task?
> Whereas the task classification accurately discriminates between the different tasks we observe that the task classification is more challenged separating the types of stimuli within a task, i.e. LRP contralateral vs. Ipsilateral and P3 rare vs. P3 frequent are difficult to separate whereas the different paradigms, i.e., LRP, P3 etc.  generally differentiates well from the other stimuli tasks. The classification performance is evaluated on not only correctly predicting the paradigm but also the two tasks within each stimuli paradigm correct, see also confusion matrices in the Supplementary material section F.2. This is the reason why the classification performance for task is in general lower than for subjects.
> ### Re. Figure 3A reveals that SLP-AE exhibits a topologically similar embedding, whereas C-AE and CSLP-AE do not demonstrate a similar pattern. It would be beneficial to provide additional explanation for this discrepancy.
> We attribute the similar topology of SLP-AE to the SLP-AE's ability to handle identical encodings in the subject and task spaces by the decoder learning to read out the subject and task information separately from the two spaces. To alleviate this issue, we propose the use of contrastive learning and note that this issue could also be alleviated with careful bottleneck tuning which in general is much more challenging to adequately tune than imposing contrastive learning to promote subject and task disentangling respectively in the two spaces. This is discussed in lines 144-152 of the paper. In the supplementary, we further consider a quadruplet latent permutation framework in which structural information cannot be preserved in the decoder and observe that this prevents the identical latent representations observed when preserving structural information in the split-latent permutation framework.
> ### Re. Additionally, the t-SNE embeddings of T. space from C-AE and CSLP-AE show similarity, despite the author's claim that Split-Latent Permutation is a key contribution. Further clarification on this aspect would be helpful.
> As the CSLP-AE includes contrastive learning we can expect some similarity between the two. However, from the quantitative evaluations of performance we observe that CSLP-AE has improved subject and task discrimination from the extracted latent spaces. These effects are consistently observed also on the movement/imagery dataset [x] provided in the Author Rebuttal.
> ### References: See Author Rebuttal.

---

> > ### Author Response · Authors · 2023-08-10
> > **References [xiv] and [xv] in the rebuttal**
> >
> > [xiv] Chorowski, et al. IEEE/ACM Trans. Audio, Speech, Language Process., 2019, doi: 10.1109/TASLP.2019.2938863.
> >
> > [xv] van den Oord, et al., 2019, arXiv:1609.03499 [cs.SD].

---

> > ### Comment · Reviewer_qodj · 2023-08-16
> >
> > Thanks for the authors' responses and clarification. I have upgraded the Rating to Weak accept

---

### Official Review · Reviewer_VVQF · 2023-07-06

**Soundness:** 3 good
**Presentation:** 3 good
**Contribution:** 2 fair
**Rating:** 6
**Confidence:** 3

**Summary:**

The paper proposes the use of swappable auto-encoder embedding along with contrastive learning in the embedding space to improve the performance of subject and task classification on evoked potentials in EEGs beyond using just an auto-encoder with contrastive learning in the embedding space. The supervised performance is improved and zero-shot conversion of EEG to new subjects or tasks is evaluated on real data from 40 subjects.

**Strengths:**

The paper is well written and clear.

The results are on large dataset (for EEG)  show consistent improvements.

The discussion are extensive and insightful. The paper shows that alone split latent permutation does not promote disentanglement and contrastive learning is needed to encourage disentanglement.  This can be seen by comparing  the task-on-subject and subject-on-task to the task and subject classification, respectively, SLP-AE is actually higher on the former two and the gap for CSLP-AE is much less than the gap for the C-AE.


**Weaknesses:**

My main concern is the lack of discussion on the generality of this method for EEG beyond ERPs. It should clarify that the current approach is likely limited to ERP and other short task responses in comparison to responses that are not time locked or have more variation in phase and duration than ERPs (e.g., sleep stage classification or seizure early warning for epilepsy subjects). It is not clear to me how this approach would be applied to more continuous cases without time-locked responses in relatively short windows. Insights from the prior work reference on speech conversion are welcomed.

While a number of ablations and baselines are tested, the papers results would be more significant if they were compared to the related work [4,37,42].

The methodology is quite expansive in terms of hyperparameter selection, but the fixing of the relative loss terms that may have vastly different units and scales (e.g., cross-entropy and MSE for instance) may not be optimal. I'd like to see more discussion on how these loss terms may interact during learning if they differ in scale. In particular it seems that C-AE could be made to perform better if perhaps the contrastive loss was weighted more heavily than the auto-encoder. The split permutation reconstruction provides some similar information in the same rage as the reconstruction error.  While likely out of scope it would be interesting to see how the proposed approach would work on simple image datasets with synthetic structure like a colored version of MNIST where the color substitutes for a subject.

**Questions:**

Comment 1: Line 100, the statement of the choice of using a shared encoder, "presently employ a shared encoder to reduce the number of parameters used", fails to acknowledge that the style embedding may be more efficiently performed using different processing architecture.

Comment 2: A mix of capitals and lowercase for subject index is used in Figure 2, which doesn't match the text. I think there should be hats on the variables on the output boxes on right side.

Question 1: In Figure 1, what are $k$ and $p$?

Question 2: How many attention heads are in the transformer?

Question 3:  Is it necessary to use non-linear classifiers in the split latent space? In much work on contrastive learning a linear classifier is used to probe the information content of the latent space. Is the use of a non-linear classifier necessary because of the complexity of the transformer in the decoder?

Comment 3: The proposed work seems to be a general approach (not specific to EEG). Notably, the papers approach is introduced with insights from speaker and speech conversion that used a fixed encoder for speaker. The concept of swapped latent permutations and contrast learning could be used more generally including with a pretrained encoder for either subject or task while training the other encoder and the shared decoder.  This possibility and any key difference should be discussed to round out the initial motivation.

Comment 4: In a similar vein to using a pretrained encoder, staggering the training of the encoders by freezing and unfreezing each of them such that one acts as conditional input is perhaps one way to disentangle the latent spaces of SLP-AE without contrastive learning. Any thoughts?

Comment 5: Figure 1 could be expanded as it is quite dense and has a relatively small font size.


**Limitations:**

The limitations are mentioned in the weaknesses.

---

> ### Author Rebuttal · Authors · 2023-08-10
>
> We thank the reviewer for the constructive comments and careful assessment finding that the paper is well written and clear, results show consistent improvements on a large dataset (for EEG), and that the discussions are both extensive and insightful.
> ### Re. Generality beyond ERPs.
> We include two additional widely used benchmark datasets for EEG analyses: the  motor imagery (MI) dataset [x, iii], and the Sleep-EDF Expanded (SleepEDFx) database [v, iii]. For the MI dataset, we consider 3 s post-stimuli EEG for MI classification. Although MI is time-locked like ERPCore, the 3 s period reflects response time which can be considered less time-locked to stimuli. The SleepEDFx database was used to further investigate the generality of the proposed approach to non-stimuli-locked data as suggested.
> We observe similar comparative performances as the ERPCore dataset on the MI, namely that the CSLP-AE procedure provides superior performance to C-AE and SLP-AE. We further contrast our approach to state-of-the-art (SOTA) procedures for the same datasets namely [i], [ii], [vi], [vii], [viii], [ix], [xi], [xii] and [xiii], and find that the performance of the proposed CSLP-AE is on-par with these current SOTA procedures for MI.
> ### Re. Comparison to related work.
> We did not include comparisons to [4,37,42] as the classification setup in [4] is based on binary classification using samples of same task vs. different task as opposed to the presently considered multi-task classification procedure.
> [37] and [42] did not produce explicit subject latent representations but used adversarial training to achieve latent task representations that are subject invariant. This is a slightly different aim than ours, which is to explicitly provide both subject and tasks latent representations which are further used in zero-shot conversion.
> ### Re. Loss terms.
> To keep the ablations simple and the computational overhead of model estimation reasonable we did not carefully tune regularization factors but fixed these at equal weight. Notably, loss magnitudes of the different components were on the same scale and with this setting of identical regularization the effect of the loss contributions had the anticipated impact on the generalization results. We have included in the PDF examples of the loss contributions highlighting their similar scale. We did not investigate careful tuning beyond the ablations systematically setting loss terms to zero.
> ### Re. Comment 1: Line 100, the statement of the choice of using a shared encoder.
> We will include this in the paper noting that other choices of architectures could be relevant also to explore such as a separate as opposed to shared encoder. We used a shared encoder to reduce the number of parameters. However, there is a small part of the bottleneck which is separate for each latent space. The encoder can thereby learn general patterns of the input and then separately learn specific encodings as part of the per-latent space separate ConvBlocks.  We do not reject the possibility of using separate encoders for both latent spaces and view this as an expansion of the framework.
> ### Re. Comment 2: A mix of capitals and lowercase for subject index.
> We apologize for the confusion. We will fix the figure to match the notation of the text.
> ### Re. Question 1: In Figure 1, what are $k$ and $p$?
> $k$ is kernel size and $p$ is padding. We used zero-padding of 1 on each side of the signal when doing convolutions with a kernel size of 3 to have equal output. This will be clarified in the figure text.
> ### Re. Question 2: How many attention heads are in the transformer?
> The transformer models used a single fully-connected attention head per layer. We opted for this due to memory requirements of using more heads per layer rather than one single large head.
> ### Re. Question 3: Is it necessary to use non-linear classifiers in the split latent space?
> There is no guarantee that the latent space will be linearly separable due to the the reconstruction and latent permutation losses. We therefore used a standard XGBoost  procedure which is a non-linear classification procedure that is widely used and found to perform well in practice. To keep the analysis unbiased by choices of classification procedures we only considered this classifier for all the considered procedures and model ablations. Results from simpler classification procedures are provided in Table 17 of Supplementary Materials section G.
> ### Re. Comment 3: Use of pre-trained encoders
> Pretrained encoders have indeed been used successfully in the context of voice-conversion, c.f. [40]. However, a drawback of such pretrained encoders is that the resulting latent space from the non-fixed encoder relies on the fixed conditionings of the pre-trained encoder. The two encoders cannot then co-adapt and adjust their latent representations with respect to each other during training. Whereas we agree that such pretrained encoders could have been employed, our proposed approach of training both encoders simultaneously provides a simple procedure for learning, which we find more elegant than pretraining one of the encoders. However, it would be interesting in future work to explore if such pre-trained encoders can enhance performance potentially also by initializing the respective encoders from pretrained models tuned to characterize subject and task latent representations, respectively.  We did not find in practice challenges minimizing the complete loss, see also the included training curves in the Author Rebuttal PDF.
> ### Re. Comment 4: Training the encoders by freezing and unfreezing each of them.
> See answer to previous question.
> ### Re. Comment 5: Figure 1 could be expanded as it is quite dense and has a relatively small font size.
> We will increase the size of the figure including font sizes in the revised manuscript.
> ### References: See Author Rebuttal.

---

> > ### Comment · Reviewer_VVQF · 2023-08-14
> >
> > **Re. Generality**
> > Thank you for performing the experiments. It is good to see that MI performance is on par.  I think that the drop in performance for sleep staging compared to supervised learning is noteworthy and should be discussed. As there were no visualizations, I'm not completely sure what to expect from the reconstruction to learn when the response is not time locked. In the case of the sleep data, is the time domain completely replaced by a spectral representation? This wasn't so clear in the rebuttal. If so the shift-invariance won't matter but a PSD estimate may lose significant amount of information compared to the time-domain signal.  This could explain the drop in performance. I would like to see/read more discussion from the authors that clearly delineates the scope where the method is expected to perform well.
> >
> > **Re. Comparison to related work.**
> > I understand that the baseline works are distinct methodologies, but this doesn't prevent comparisons in performance. Couldn't the methods that are trained to be invariant to subject be applied here?

---

> > > ### Author Response · Authors · 2023-08-15
> > >
> > > We thank the reviewer for responding to our rebuttal for clarifications.
> > >
> > > ### Re. Generality
> > > We will include a further discussion of the drop in performance and limitations.
> > >
> > > We have employed the conventional 30 s time series windows with associated sleep stage labels as common in the sleep stage literature (as well as being the conventional clinical procedure). The single channel EEG time information sampled at 100 Hz is transformed using a 128-point short-time-Fourier transform with a 100-point window size and 15-point step size resulting in a 64x200 frequency x time representation. We believe this preserves relevant temporal information while allowing sufficient frequency-domain precision, but indeed it may be that working in the raw time domain or applying intermediate filters could improve performance.
> > >
> > > We believe the CSLP-AE framework works in a non-time-locked setting but agree with the reviewer that this is a much more challenging setting. We attribute the ability of CSLP-AE to also work when data is not time-locked to the autoencoder loss reconstructing from same sample to same sample. Thereby, the encoder can encode subject, task and structural information specific to the given sample and use this structural information in the reconstruction. This can admit information such as timing of the response etc. to be preserved in the reconstruction. However, the quadruplet split latent permutation (QSLP-AE) framework considered in the supplementary material blocks all structural information by reconstructing samples from different samples. As opposed to the CSLP-AE we therefore expect the QSLP-AE to be unable to work when data is not time-locked. On the other hand, the QSLP-AE can achieve (D.s, D.t) conversion (i.e., conversion between different speaker and task at the same time) which is challenging and a current limitation of the CSLP-AE framework.
> > >
> > > ### Re. Comparison to related work.
> > > In the limited time of the rebuttal, we focused on data diversity and state-of-the-art deep learning procedures by including two new open-access datasets in which SOTA was available (the dataset considered in [4] is proprietary, while the data in [42] requires administrator approval of up to a week).  Apart from not finding [4,42] directly applicable in their considered setups as explained in the rebuttal, no code is currently available for the [4, 42] procedures to the best of our knowledge. Including [4, 42] would therefore require reimplementing these procedures from scratch and tuning these models to the current datasets. While we agree this could potentially be done, this runs the risk of reimplementation issues potentially biasing results, which is mitigated by benchmarking directly on the two widely used datasets included to compare with SOTA.
> > >
> > > In [37] they show that EEGNet [xi] performs best on task classification but does not benefit from adversarial training compared to supervised training. In the Author Rebuttal we included results from EEGNet using supervised training to which we explicitly compare against.

---

### Author Rebuttal · Authors · 2023-08-09

We thank the reviewers for their efforts in reviewing this manuscript and providing constructive and insightful comments. Individual responses to each reviewer’s questions and comments have been prepared and are provided below where appropriate.

Multiple reviewers address the fact that results have only been provided for a single dataset (ERPCore). While the dataset does contain several different experimental paradigms, we agree that focusing on only ERP-related data might be too narrow in scope. We have therefore trained and tested our framework on two additional benchmark EEG datasets: the EEG Motor Movement/Imagery Dataset (EEGMMI) [x, iii] containing EEG recordings from 109 subjects performing several different motor imagery (MI) tasks; and the Sleep-EDF Expanded (SleepEDFx) database [v, iii] containing 153 single-EEG polysomnography studies from 78 unique subjects. The SleepEDFx is included to explicitly probe the model integrity on data that is not time-locked to an external stimulus, as opposed to the ERPCore dataset.

As shown in the Author Rebuttal associated PDF, our framework can accurately classify MIs at a level that is on par with current state-of-the-art (64.28 ± 0.16 % for our model vs. 65.07% for an EEGNet-based model [xi], see Table 1 in the Author Rebuttal associated PDF). We also find that our framework generalizes well to EEG data that is not time-locked to an external stimulus. As shown in Table 2 in the Author Rebuttal associated PDF, our framework can predict sleep stages (Wake, REM, Non-REM 1, 2, 3) at a level of 75.16±0.96 %% accuracy with minimal changes to the framework.

We applied our framework “as is” with the following minor modifications: since the SleepEDFx dataset contains a limited number of EEG channels, we only considered a single EEG channel and applied a short-time Fourier transform to fit the data to our setup. Furthermore, to match the latent dimension of the models from the present study, we opted to only use three blocks instead of four and resized the latent dimension’s temporal resolution from 64 to 40 to end up with latents of size 25x40 compared to the present study which had a 16x64 latents. Finally, these models were only run for 50 epochs due to time constraints and therefore not to convergence. Detailed information regarding experimental setups for the two additional datasets will be provided in the updated supplementary material.

For the EEGMMI dataset the samples had 3 seconds duration at a sampling rate of 160 Hz, and we therefore needed to use five blocks instead of four to match the latent dimension of the present study. The models were run to convergence for 800 epochs. No other tweaks were made to conform to this dataset.

While these results on additional databases are on par but do not exceed state-of-the-art, they are trained "as is”, employing the architecture, most hyperparameters and framework as used in the present study. Additionally, they are based on learning self-supervised latent representations as opposed to the fully supervised, current state-of-the-art classification models. Furthermore, no significant additional model optimization via excessive hyper-parameter tuning was attempted here, which may or may not yield additional improvements in task accuracy on all datasets considered.

To keep the ablations simple and the computational overhead of model estimation reasonable, we did not carefully tune regularization factors but fixed these at one (i.e., equal weighting for all losses). For such equal weighting to work properly, the magnitudes of the different loss components need to be on the same, or similar, scale. We have provided in the Author Rebuttal associated PDF examples of the different loss contributions highlighting their similar scale ensuring the validity of using equal weighting for the losses. Future work should prioritize further hyperparameter optimization.

# References

[i] Dose et al., Expert Syst. Appl., 2018, doi: 10.1016/j.eswa.2018.08.031.

[ii] Eldele et al., IEEE Trans. Neural Syst. Rehabil. Eng., 2021, doi: 10.1109/TNSRE.2021.3076234.

[iii] Goldberger et al., Circulation, 2000, doi: 10.1161/01.CIR.101.23.e215.

[iv] Kappenman, et al., NeuroImage, 2021, doi: 10.1016/j.neuroimage.2020.117465.

[v] Kemp, et al., IEEE Trans. Biomed. Eng., 2000, doi: 10.1109/10.867928.

[vi] Mousavi, et al., PLOS ONE, 2019, doi: 10.1371/journal.pone.0216456.

[vii] Phan, et al., IEEE Trans. Neural Syst. Rehabil. Eng., 2019, doi: 10.1109/TNSRE.2019.2896659.

[viii] Phan, et al., IEEE Trans. Pattern Anal. Mach. Intell., 2022, doi: 10.1109/TPAMI.2021.3070057.

[ix] Phan, et al., IEEE Trans. Biomed. Eng., 2022, doi: 10.1109/TBME.2022.3147187.

[x] Schalk, et al., IEEE Trans. Biomed. Eng., 2004, doi: 10.1109/TBME.2004.827072.

[xi] Wang, et al., IEEE MeMeA, 2020. doi: 10.1109/MeMeA49120.2020.9137134.

[xii] Xie et al., IEEE Trans. Neural Syst. Rehabil. Eng., 2022, doi: 10.1109/TNSRE.2022.3194600.

[xiii] Zhu, et al. Int. J. Environ. Res. Public Health, 2020, doi: 10.3390/ijerph17114152.

[4] Bollens et al., ICASSP, 2022. doi: 10.1109/ICASSP43922.2022.9747297.

[37] Özdenizci et al., IEEE Access, 2020. doi: 10.1109/ACCESS.2020.2971600.

[40] Qian et al., ICML, 2019. arXiv:1905.05879 [eess.AS].

[42] Rayatdoost et al., ICASSP, 2021. doi: 10.1109/ICASSP39728.2021.9414496.

---

### Decision · Program_Chairs · 2023-09-21

**Decision:**

Accept (poster)

**Comment:**

All reviewers have found merit in this work and judged it clear, meaningful and even insightful. Constructive interactions happened between reviewers and authors during discussion period. This work should be endorsed for publication at NeurIPS 2023.